# Automated phenotyping of postoperative delirium-like behaviour in mice reveals the therapeutic efficacy of dexmedetomidine

Silu Cao[1,2,3,4,8], Yiling Wu[5,8], Zilong Gao[6,7,8], Jinxuan Tang[1,2,3,4], Lize Xiong [1,2,3,4], Ji Hu [5✉] & Cheng Li [1,2,3,4✉]

Postoperative delirium (POD) is a complicated and harmful clinical syndrome. Traditional behaviour analysis mostly focuses on static parameters. However, animal behaviour is a bottom-up and hierarchical organizational structure composed of time-varying posture dynamics. Spontaneous and task-driven behaviours are used to conduct comprehensive profiling of behavioural data of various aspects of model animals. A machine-learning based method is used to assess the effect of dexmedetomidine. Fourteen statistically different spontaneous behaviours are used to distinguish the non-POD group from the POD group. In the task-driven behaviour, the non-POD group has greater deep versus shallow investigation preference, with no significant preference in the POD group. Hyperactive and hypoactive subtypes can be distinguished through pose evaluation. Dexmedetomidine at a dose of $25\,\mu g\,kg^{-1}$ reduces the severity and incidence of POD. Here we propose a multi-scaled clustering analysis framework that includes pose, behaviour and action sequence evaluation. This may represent the hierarchical dynamics of delirium-like behaviours.

[1] Department of Anesthesiology and Perioperative Medicine, Shanghai Fourth People's Hospital, School of Medicine, Tongji University, Shanghai 200434, China. [2] Translational Research Institute of Brain and Brain-like Intelligence, Shanghai Fourth People's Hospital, School of Medicine, Tongji University, Shanghai 200434, China. [3] Clinical Research Center for Anesthesiology and Perioperative Medicine, Tongji University, Shanghai 200434, China. [4] Shanghai Key Laboratory of Anesthesiology and Brain Functional Modulation, Shanghai 200434, China. [5] School of Life Sciences and Technology, ShanghaiTech University, Shanghai 201210, China. [6] School of Life Sciences and Key Laboratory of Growth Regulation and Translational Research of Zhejiang Province, Westlake University, Hangzhou 310024, China. [7] Present address: Chinese Institute for Brain Research, Beijing 102206, China. [8] These authors contributed equally: Silu Cao, Yiling Wu, Zilong Gao. ✉email: huji@shanghaitech.edu.cn; chengli_2017@tongji.edu.cn

Postoperative delirium (POD) is the most common central nervous system complication after surgery in the elderly, with an incidence of 15–25%[1]. Key diagnostic features include an acute onset and fluctuating course of symptoms, inattention, impaired consciousness, and disturbance of cognition[2–5]. Patients with POD demonstrate certain characteristics including restlessness, wandering, fear, increased speech, alertness, apathy, withdrawal[6], neurocognitive disorders[7], and attention deficits[8]. POD is divided into two clinical subtypes based on the appearance of two clinical psychomotor states: hyperactive and hypoactive. POD mainly affects elderly people by increasing the economic burden, cognitive impairment, worsening many medical illnesses, and increasing the risk of death[9,10].

However, it is difficult to conduct in-depth research to solve this challenging clinical condition. Some of the obstacles to researching the basic mechanism of POD include the lack of accurate behavioural detection of delirium in animal models, the inability to sensitively detect animals with POD, and the inability to accurately determine its subtypes. Previous studies have examined delirium-like behaviours related to the natural and learned behaviours of rodents[11,12]. The elevated plus maze (EPM) and open field test (OFT) are mainly used for anxiety assessment[13,14]. The classical anxiety parameter is the time spent in a given area of the behavioural device in a fixed period. Maze experiments and novel object recognition (NOR) are accepted cognition-measuring paradigms for delirium in mice[14–17]. The primary measure of assessment was the ability of the mice to recognize the 'novel' and the 'familiar' after anaesthesia and surgery. In summary, previous methods mainly evaluate animal behaviours by constraining the animal to certain parameters and evaluating animal performance.

Animal behaviour is driven by neural activity. The hierarchical, multiscale neural activity also corresponds to hierarchical, multiscale behaviour. Behavioural changes caused by neural circuit abnormalities can be reflected in all levels of behaviours, such as short-scale kinematic abnormalities related to posture and long-scale behaviour dynamic abnormalities related to action sequence (AcSeq). Notably, animal behaviour is a bottom-up and hierarchical organizational structure composed of time-varying posture dynamics[18–20]. Previous theories[21] and recent data analysis[20] have shown that animal behaviour is like the "letter-word-sentence" of human language. Spontaneous behaviour has traditionally been perceived as being intrinsically stable[22]; and in past reports, it has been used to distinguish transgenic animals[23,24] and describe disease states[25–27]. Furthermore, task-related behaviours also change when it comes to cognitive processing. Task-driven behaviour can be understood as animal behaviour is influenced by different forms of behavioural contexts, such as spatial attention, object-based attention, and feature-based attention, as well as task-dependent and anticipative effects[28–31]. These interactions are context-modulated by the animal's state of sustained attention[28]. Variability in internal brain states related to cognitive variables (e.g., attention, alertness, task engagement, and arousal) affect visual encoding and perception[32]. Therefore, in addition to assessing the behaviour in the constrained paradigm, it is more important to comprehensively evaluate spontaneous and task-driven behaviour.

The latest advances in deep learning have allowed for accurate, fast, and robust measurements of animal behaviour. Moreover, in the past few years, many powerful tools have been specifically designed to help measure pose and estimate behaviour[24,33–35]. Recently, these tools have been used to evaluate diseases such as anxiety and defensive behaviour[36,37]. However, behavioural analysis strategies for complex syndromes, such as POD which is characterized by different subtypes with very different manifestations, are still lacking. Additionally, there have been a few reports of dexmedetomidine (Dex) acting on POD[38–41]. Moreover, the mechanism of the neuroprotective effect of Dex has been given more attention[42]. However, the efficacy of Dex has not been verified in detail or in animal models.

To address these obstacles, in this study we proposed a multiscaled clustering analysis framework that includes pose, behaviour and AcSeq with spontaneous behaviour and task-driven behaviour to estimate POD and its subtypes. First, we used anaesthesia and surgery to establish the mouse model. We established that the POD group that underwent anaesthesia and surgery had abnormal behavioural patterns. Therefore, we differentiated the POD and non-POD groups in the model by unsupervised clustering of spontaneous behaviours. Second, for the task-driven behaviour, we used AcSeq to assess the cognitive level of the POD mice to reveal their exploratory behavioural preferences. Third, we introduced an assessment of pose parameters that accurately captured the highly significant symptoms of the two subtypes with opposite kinematics in the animal models. Finally, using machine learning-based methods together with the further effective analysis of behavioural monitoring under Dex treatment, we demonstrate that our strategy confirms the important role of Dex in reducing the severity and incidence of POD in an animal model.

## Results

**Collecting mouse multi-scale data with a 3d multi-view motion capture system.** First, we performed abdominal surgery under isoflurane anaesthesia to induce a delirium-like behaviour in the mice (Fig. 1a). The group that underwent the anaesthesia and surgery was named the model group, and the group without anaesthesia and surgery was named the control group. Then, we captured the delirium-like behavioural changes in multi-scale patterns at 6, 30, 54, and 78 h in a featureless circular OFT and NOR using a multi-view video capture system[24] (Fig. 1b, S1). The OFT was used to detect spontaneous behaviour. The NOR was used to detect the task-driven behaviour.

All processes are described in Fig. 1c. Pipeline of mouse behaviour recording and analysis via 3D-motion learning framework. Top: 3 steps for single mouse 3D body reconstruction and movement segmentation. First, using pre-trained DeepLabCut model to track the 2D coordinates of 16 body parts from each camera view; second, reconstructing the mouse 3D skeleton by fusing four views 2D coordinates; third, using BehaviorAtlas (BeA, see Methods) to decompose and segment mouse movements. Bottom: merging all the movement segments of involved mice and constructing the Similarity Kernel Matrix. Then using a dimensional reduction algorithm Uniform Manifold Approximation and Projection and hierarchical clustering for movement clustering. Finally, 40 types of movement were categorized for further analysis of POD mice.

**Delirium-like behaviour is accurately described with spontaneous behaviour in the model group.** To classify the POD and non-POD mice in the model group, the behavioural pattern of the spontaneous behaviour was divided into two clusters: cluster1 and cluster2 (Fig. 2a). Based on the assumption that POD mice were from the model group, we identified the POD group (the most different from the control group) and the non-POD group (the most similar to the control group) (Fig. 2b, S2a). The traditional diagnosis of POD usually used the static parameter composite Z score[13] (see Methods). Using this approach, we found significant differences between the control group and the model group (Fig. S2b). However, using this method, no significant differences were found between the POD and non-POD groups (Fig. S2c). Therefore, by comparing the behaviour between the two groups,

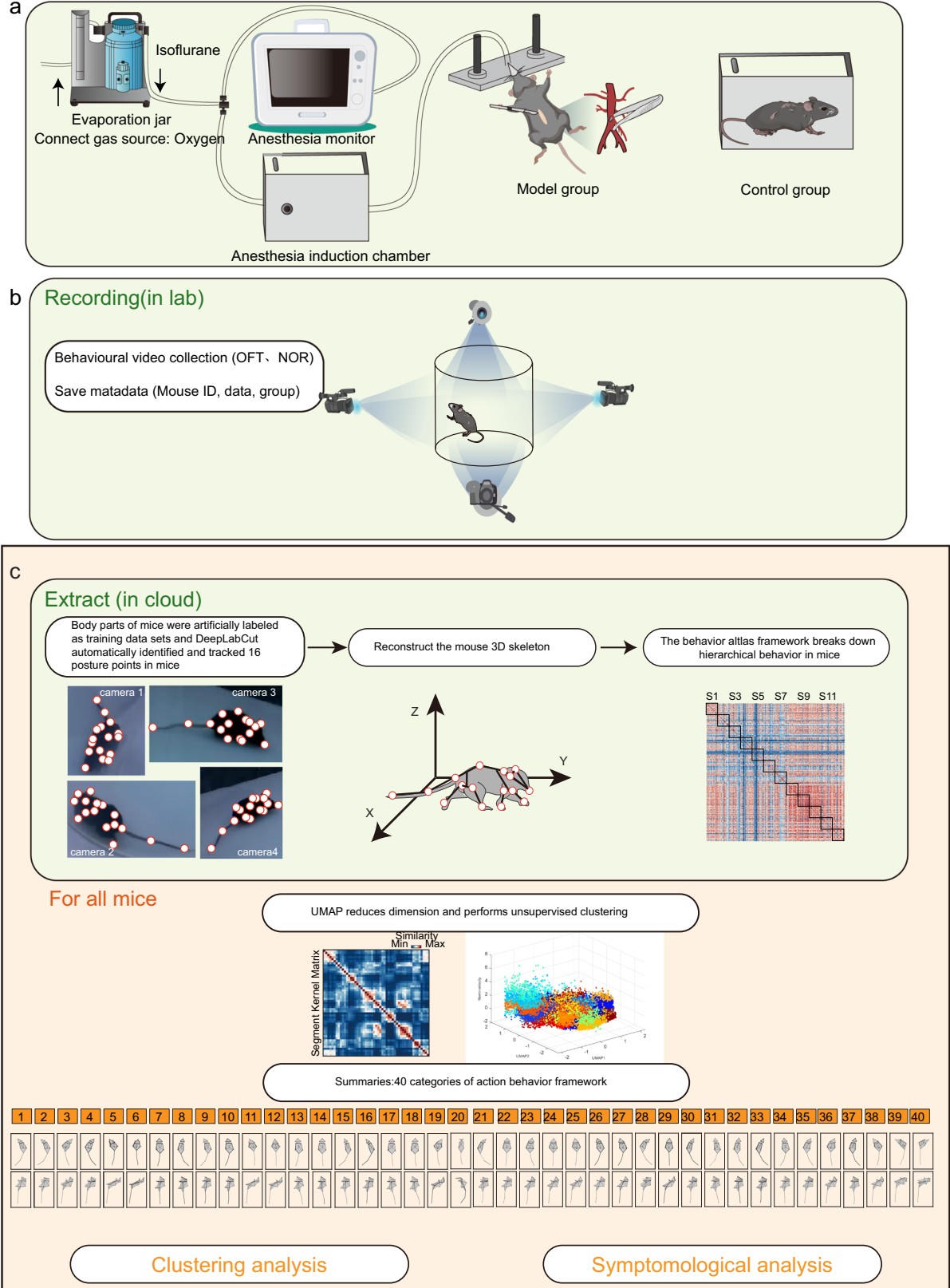

**Fig. 1 Collecting mouse multi-scale data with a 3D multi-view motion capture system. a** Flow chart of exploratory laparotomy. Model group: undergo the anaesthesia and exploratory laparotomy; control group: no anaesthesia or surgery. **b** Behavioural recordings under multi-view cameras in freely moving aged mice. **c** Flow charts for obtaining 'pose' and 'behaviour' parameters. Top: 3D body reconstruction and movement segmentation for a single mouse. Bottom: Movement clustering and phenotyping for all mice. For specific methods, refer to the ref. [24].

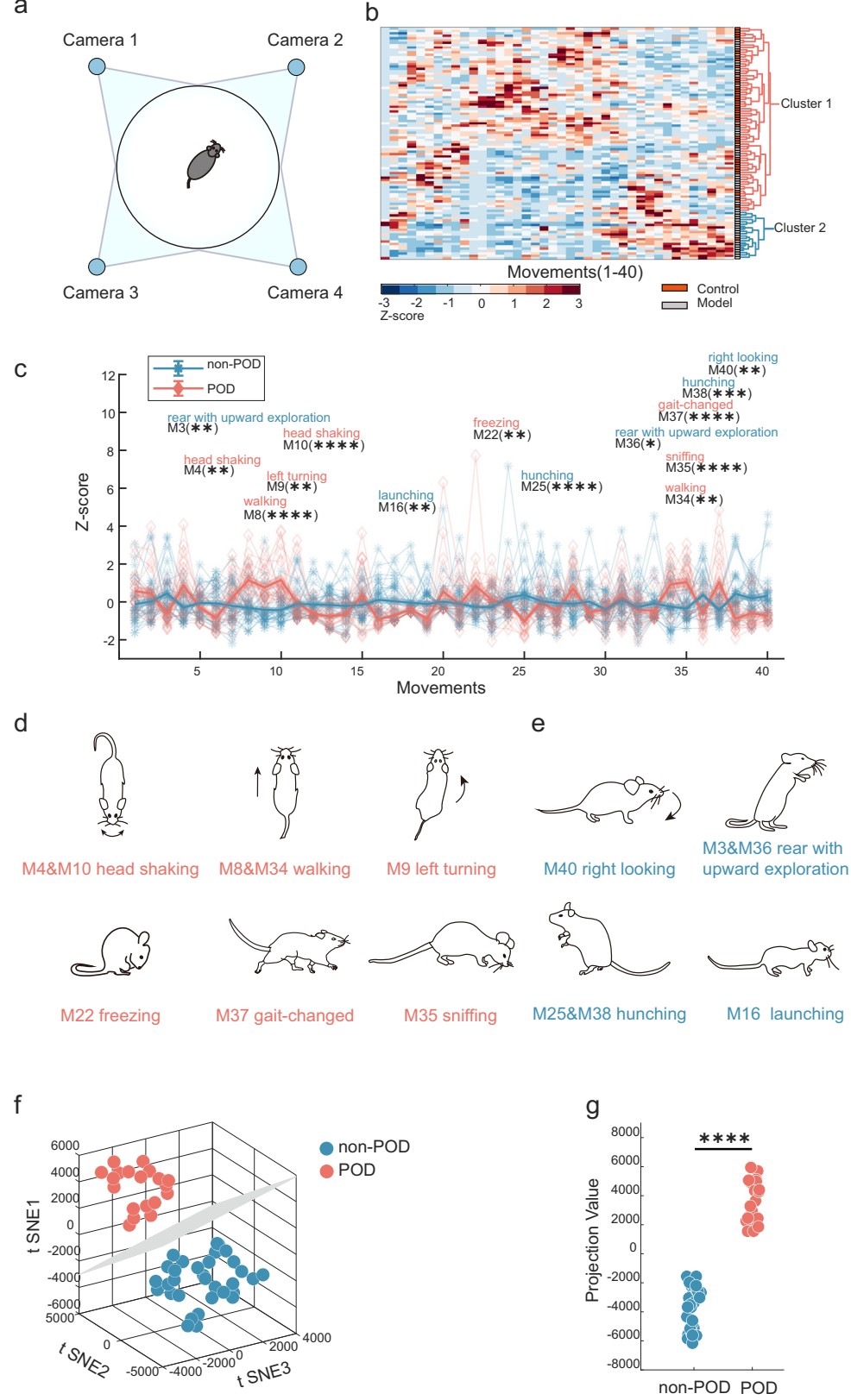

delirium-like behaviour was found in previously unreported ways. There were 14 movements with significant differences between the POD and non-POD groups (Fig. 2c). POD mice spent a much higher percentage of time exhibiting eight of these behaviours (Fig. 2d). By manually reviewing these eight types of video clips, we annotated M4 and M10 showing head shaking, M8 and M34

associated with walking, M9 indicates turning left, M22 indicates freezing, and M35 indicates sniffing. In addition, M37 indicates gait-change. Next, a significant reduction was observed in six movements (Fig. 2e), M3 and M36 (rear with exploration) and M40 (right-looking). Likewise, M25 and M38 indicate hunching. M16 (launching) appears as a sudden acceleration.

**Fig. 2 POD was diagnosed by unsupervised clustering and low-dimensional embedding based on spontaneous behaviours. a** Schematic diagram of open field behaviour under multi-angle shooting. **b** Clustergram of samples with 40 spontaneous behaviour modules (grey, model group, $n = 52$; orange, control group, $n = 46$). **c** Comparison of the proportion of behaviour types between non-POD and POD mice. The bold traces and shadows indicate the mean ± SEM. Fractions of each group and light colour traces are the fractions of all 52 samples (blue, non-POD, $n = 33$; pink, POD, $n = 19$). Fourteen movements showed significant differences between the two groups. M4 (non-POD $= -0.263 ± 0.12$, POD $= 0.907 ± 0.31$), M10 (non-POD $= -0.389 ± 0.115$, POD $= 1.179 ± 0.287$), M8 (non-POD $= -0.276 ± 0.140$, POD $= 1.168 ± 0.241$), M34 (non-POD $= -0.216 ± 0.181$, POD $= 0.947 ± 0.237$), M9 (non-POD $= -0.386 ± 0.168$, POD $= 0.792 ± 0.213$), M22 (non-POD $= -0.237 ± 0.094$, POD $= 0.886 ± 0.415$), M37 (non-POD $= -0.405 ± 0.092$, POD $= 1.077 ± 0.344$), M35 (non-POD $= -0.321 ± 0.131$, POD $= 1.074 ± 0.256$), M3 (non-POD $= 0.477 ± 0.204$, POD $= -0.574 ± 0.161$), ; M36 (non-POD $= 0.410 ± 0.194$, POD $= -0.559 ± 0.189$), M40 (non-POD $= 0.356 ± 0.224$, POD $= -0.694 ± 0.097$), M25 (non-POD $= 0.408 ± 0.186$, POD $= -0.985 ± 0.129$), M38 (non-POD $= 0.457 ± 0.217$, POD $= -0.849 ± 0.124$), M16 (non-POD $= 0.141 ± 0.168$, POD $= -0.896 ± 0.088$). Statistics: two-way ANOVA followed by the Sidak post hoc multiple comparison test, **M3, $P = 0.0061$; **M4, $P = 0.0012$; ****M8, $P < 0.0001$; **M9, $P = 0.0011$; ****M10, $P < 0.0001$; **M16, $P = 0.0069$; **M22, $P = 0.0022$; ****M25, $P < 0.0001$; **M34, $P = 0.0013$; ****M35, $P < 0.0001$; *M36, $P = 0.0164$; ****M37, $P < 0.0001$; ***M38, $P = 0.0001$; **M40, $P = 0.0061$. **d** Schematic diagram of a series of behaviours that increase in the POD group. **e** Schematic diagram of a series of behaviours that decrease in the POD group. **f** Low-dimensional representation of the two animal groups (blue, non-POD, $n = 33$; pink, POD, $n = 19$). The 52 samples in 3D space were dimensionally reduced from 14-dimensional movement fractions, and they are well separated. The grey plane is the decision boundary in the 3D space of the two groups. **g** The projection values were calculated by projecting the points in (**f**) onto the decision boundary (pink, POD $= 3636 ± 332.8$, blue, non-POD $= -3793 ± 250.2$; ****$P < 0.0001$, Mann–Whitney test). All data were presented as mean ± SEM.

Next, fourteen significantly different behavioural modules were combined with low-dimensional embeddings for further analysis[24]. First, a bioinformatic approach was used followed by using t-distributed stochastic neighbour embedding (t-SNE) to reduce the 14 dimensions of the movement modules in 3D space (Fig. 2f, see Methods). A linear classifier can completely classify the two groups, demonstrating that t-SNE can represent high-dimensional information of behaviour in a 3D space. Next, the projection values of 52 data points to a linear classifier in 3D space were calculated, and a one-dimensional (1D) representation was obtained (Fig. 2g). A significant difference was observed between the two groups with no overlapping data points. Thus, our approach can detect the differences observed through spontaneous behaviour paradigms.

**Detailed description of task-driven behaviour in mice with delirium.** Mice interact with objects in their surroundings for different purposes such as collecting new information to test for edibility or danger. Mice interact less with familiar objects than with novel objects[43–45]. Using the NOR test under multi-angle shooting (Fig. 3a, S3a), the behaviour of mice in the task environment[28,29] was assessed and revealed that the POD group experienced more M35 (sniffing) and less M39 (left-looking) (Fig. 3b). Next, the interaction of mice with familiar and novel objects was carefully assessed. The POD group developed less M20 (stretched attend) and more M39 (left looking) when exposed to novel objects (Fig. 3c). In addition, less M20 (stretched attend) and M31 (walking with head down) and more M40 (right-looking) occurred during contact with familiar objects in the POD group (Fig. 3d). Schematic representations of these actions are shown in Fig. 3e. Next, deep (sniff, touch, crab) and shallow (avoid) investigations were used to describe the desire and curiosity of mice for exploring objects[46] (Fig. 3f). The DSP was introduced (see Methods). The DSP is the relative time a mouse carries out a deep investigation compared with the time it spends in a shallow investigation[46]. This DSP was much greater in the novel period than in the familiar period in the non-POD group (Fig. 3g). However, no significant differences were observed in the POD group (Fig. 3h).

**The subtypes of delirium-like behaviour were revealed by clustering methods on the pose.** Based on the clinical diagnosis, delirium has two disease state phenotypes: hyperactive and hypoactive[47–49]. Classifying the syndromes of POD into different subtypes based on behavioural tendencies is important when studying the neurobiological mechanisms of POD[50]. Here, we

introduced 13 pose features, such as left front claw speed, length, nose speed, and acceleration (Fig. 4a) to describe the two groups of clinical subtypes that are kinematically opposite. To clarify the differences between the two groups, the probability mass function (PMF) of these 13 postural features was extracted (see Methods). First, using 91 eigenvalues, dimensionality reduction and unsupervised clustering were performed (Fig. 4b). The means of these 13 features from the mice of the two groups were compared (Fig. S4a–m). The results revealed consistent kinematic trends, which aligned with the characteristics of the two subtypes of POD; thus, the two groups were named Hyper and Hypo. The results revealed that the curves related to the kinematic parameters from the two groups were significantly different with a large Ks-distance (Fig. 4c–g, see Methods). In addition, some PMF curves of the five features from the two groups provided more detailed information (Fig. S4n–u). Therefore, POD mice were successfully categorized into Hyper and Hypo groups using the pose descriptions. Next, the symptomatology characteristics of the two subtypes were assessed (Fig. 4h), and the results revealed that more M4 (head shaking) and M28 (right turning) movements occurred in the hypoactive group (Fig. 4h).

**Evaluating the effects of dex in the POD animal model.** Dex is considered an effective drug to reduce POD in non-cardiac surgery[41]. However, the symptoms that can be improved have not been elucidated in animal models. Given that our experimental objective was to validate the efficacy of Dex for non-cardiac surgery, we selected a dose that has been widely reported to have neuroprotective and anti-inflammatory effects[17,51,52]. Here, we explored the effects of Dex on postoperative behaviour at a dose of 25 μg kg$^{-1}$ (Fig. 5a). First, we assessed the effect of Dex on the early improvement of spontaneous behaviour. Dex significantly improved the three movements of M2 (sniff with turning left), M6 (climbing), and M28 (right turning) (Fig. 5b). In addition, the increase in M39 (left-looking) in the POD group in the NOR can be improved (Fig. 5c), which was previously described in Fig. 3b. These may be future research directions for studying the effect of Dex on POD, especially for hyperactive subtypes. Schematic representations of these actions are shown in Fig. 5d.

The effect of Dex on POD was evaluated in an animal model using a machine learning-based method. First, we used all the mice in the POD group and the non-POD group as the training set and cross-validated the training model itself, and we then used the Training Model to label the Dex-treated group (Figs. S5a, b, see Methods). The results are represented by 3D plots in which the corresponding samples were embedded in the POD and non-

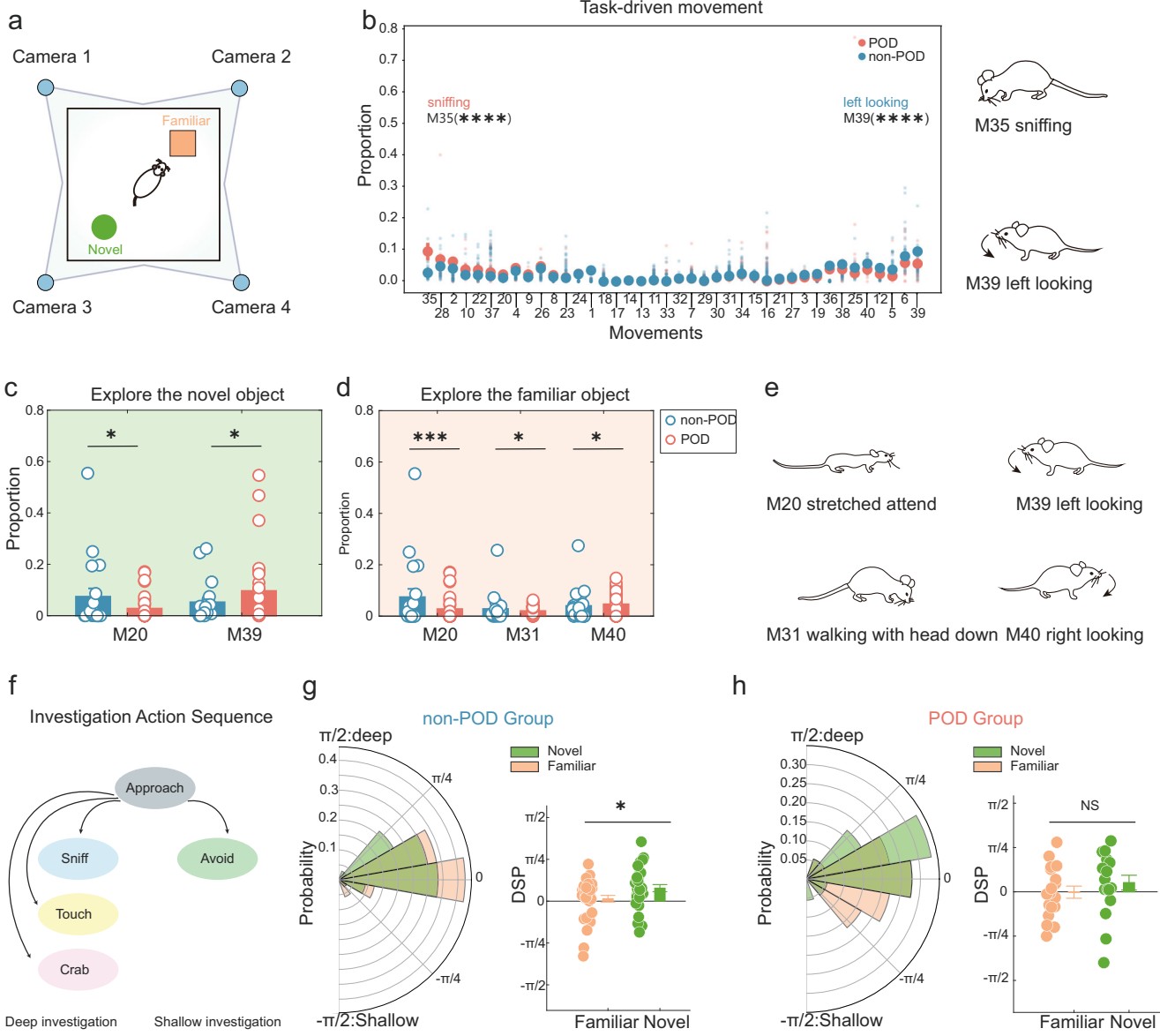

**Fig. 3 Assessment of task-driven behaviour in delirium mice. a** Schematic diagram of a novel object recognized under multi-angle shooting. Schematic diagram of the test. The orange quadrangular prism is a familiar object, which is an object that was repeatedly touched before the test. The green cone represents objects that are touched for the first time, namely new objects. For detecting the replacement of new objects in daily behaviour, see the Methods and Supplementary Figure 3. **b** Comparison of the fraction of task-driven movement types between non-POD mice and POD mice in NOR. The bold traces and shadows indicate the mean ± SEM (blue, non-POD, $n = 33$; pink, POD, $n = 19$). Two movements showed significant differences between the two groups, that POD mice prefer is sniffing (M35, non-POD = 0.028 ± 0.004, POD = 0.09 ± 0.039), less is left looking (M39, non-POD = 0.095 ± 0.011, POD = 0.053 ± 0.011) ****M35, $P < 0.0001$; ****M39, <0.0001. **c** Comparison of the fraction of movement types between non-POD mice and POD mice for exploring the novel objects. The bold traces and shadows indicate the mean ± SEM (blue, non-POD, $n = 33$; pink, POD, $n = 19$). Two movements showed significant differences between the two groups, and the fractions of the four movements that POD mice prefer are stretching ahead (M20, non-POD = 0.027 ± 0.009, POD = 0.074 ± 0.032), less is left looking (M39, non-POD = 0.096 ± 0.023, POD = 0.052 ± 0.018). *M20, $P = 0.0127$; *M39, $P = 0.0255$. **d** Comparison of the fraction of movement types between non-POD mice and POD mice for exploring a familiar object. The bold traces and shadows indicate the mean ± SEM (blue, non-POD, $n = 33$; pink, POD, $n = 19$). Three movements showed significant differences between the two groups, and the fractions of the four behaviours that POD mice prefer are stretching ahead (M20, non-POD = 0.037 ± 0.11, POD = 0.095 ± 0.032) and walking with head down (M31, non-POD = 0.017 ± 0.004, POD = 0.067 ± 0.052), less is right looking (M40, non-POD = 0.073 ± 0.018, POD = 0.028 ± 0.007). ***M20, $P = 0.0006$; *M31, $P = 0.016$; *M40, $P = 0.0374$. **e** Schematic diagram of the different behaviours. **f** We mapped the AcSeq to distinguish deep-seeking behaviour and shallow seeking behaviour in mice. **g** Probability histogram and bar graph of the DSP index of POD mice exploring familiar and novel objects. The deep versus shallow investigation preference (DSP) varies between -π/2 and π/2, where -π/2 and π/2 indicate the absolute preference for shallow and deep investigation, respectively, and 0 indicates an equal preference for deep and shallow investigation. Probability histogram and bar graph of the DSP index of POD mice($n = 19$) exploring familiar and novel objects (Familiar = −0.359 ± 5.84, Novel = 8.854 ± 7.810). Statistics: (Mann–Whitney test; NS not significant, $P = 0.323$). **h** Probability histogram and bar graph of the DSP index of non-POD mice exploring familiar and novel objects (Familiar = 2.639 ± 3.59, Novel=14.02 ± 3.84). Statistics: (Mann–Whitney test; *$P = 0.0340$). All data were presented as mean ± SEM.

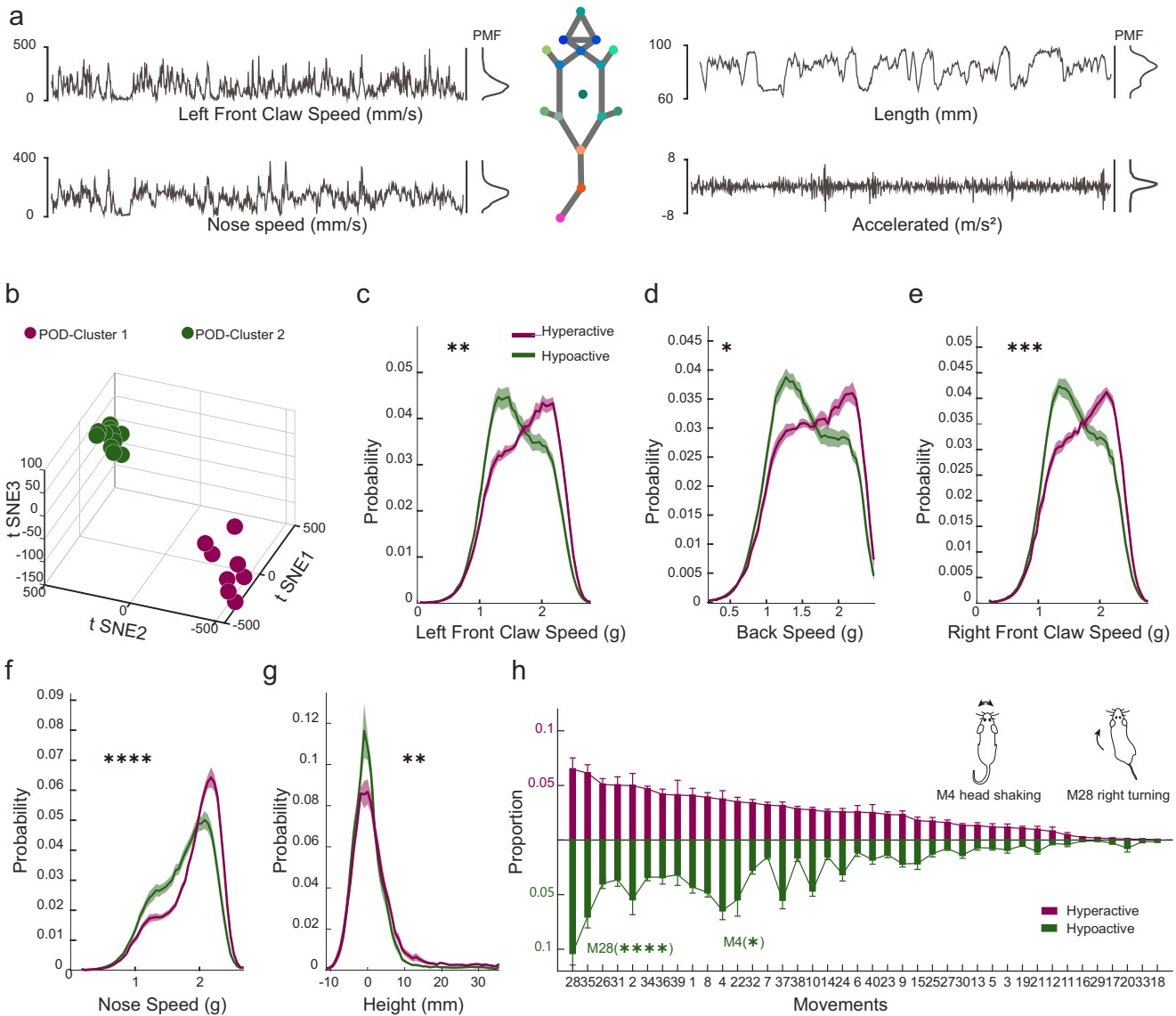

**Fig. 4 Defining two clinical subtypes of POD using posture features. a** Imaging-based distributions of the left front claw speed, length, nose speed and accelerated given as arbitrary units (a.u.) for an example mouse during a 30-s example snippet. **b** Low-dimensional representation of the two animal groups (purple, cluster 1, $n = 8$; green, cluster2, $n = 11$). The 19 dots in 3D space were dimensionally reduced from 77-dimensional movement fractions, and they are well separated with k-means. **c–g** Comparisons of PMF of certain kinematic or posture parameters for cluster1 and cluster2, including the left front claw speed (**P = 0.0047), the back speed (*P = 0.016), the right front claw speed (***P = 0.0003), the nose speed (****P < 0.0001) and the height (**P = 0.0019). Kolmogorov–Smirnov test was used was used to test for differences between the distributions of the two clusters. The bold traces and shadows indicate the mean ± SEM. Fractions of each group and light colour traces are the fractions of all 19 mice (purple, Hyper, $n = 8$; green, Hypo, $n = 11$). **h** Behaviors expressed by Hyper and Hypo mice during the OFT day of group monitoring. 40 behavioural traits were computed for each individual. The bold traces and shadows indicate the mean ± SEM. Fractions of each group and light colour traces are the fractions of all 19 mice (purple, Hyper, $n = 8$; green, Hypo, $n = 11$). Two movements have significant differences between the two groups, and the fractions of the two movements that Hypo mice prefer are right turning (M28, Hypo = 0.103 ± 0.010, Hyper= 0.065 ± 0.010) and head shaking (M4, Hypo = 0.064 ± 0.008, Hyper = 0.037 ± 0.008). Statistics: two-way ANOVA followed by Sidak post hoc multiple comparisons test, *M4, P = 0.013; ****M28, P < 0.0001. All data were presented as mean ± SEM.

POD groups (Fig. 5e, S5c). In addition, we found that the Dex-treated-POD group was closer to and significantly different from the projection value of the decision boundary than the POD group (Fig. 5f, g), suggesting that Dex has the potential to reduce the severity of delirium-like behaviours in animal models. Finally, the incidence of POD in the Dex-treated group decreased from 62% to 50% compared with those in the model group (Fig. 5h).

### Discussion
Inspired by the recent advances in computer vision, the current study presents a multi-scaled clustering analysis framework for assessing delirium-like status from pose, behaviours, and AcSeq by combining unsupervised clustering and low-dimensionality embedding. Delirium-like diagnosis was made using a clustering approach to spontaneous behaviour to classify mice, and the results fit well with the POD characteristics, disease pattern of early onset, and fluctuating trend. Assessment of task-driven behaviour indicated the presence of a cognitive decline in POD mice, and we further evaluated cognitive changes using AcSeq. Unsupervised clustering for pose parameters was used to describe the POD subtypes, and further low-dimensionality embedding was applied to represent the feature space of the subtypes. This

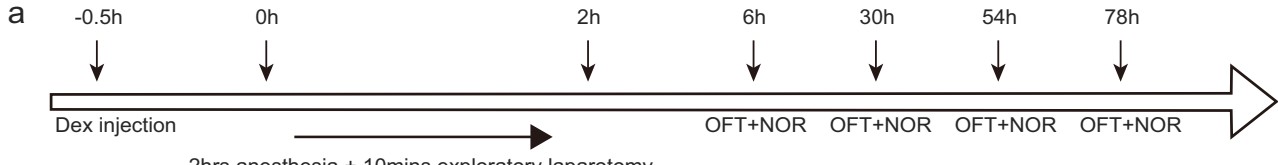

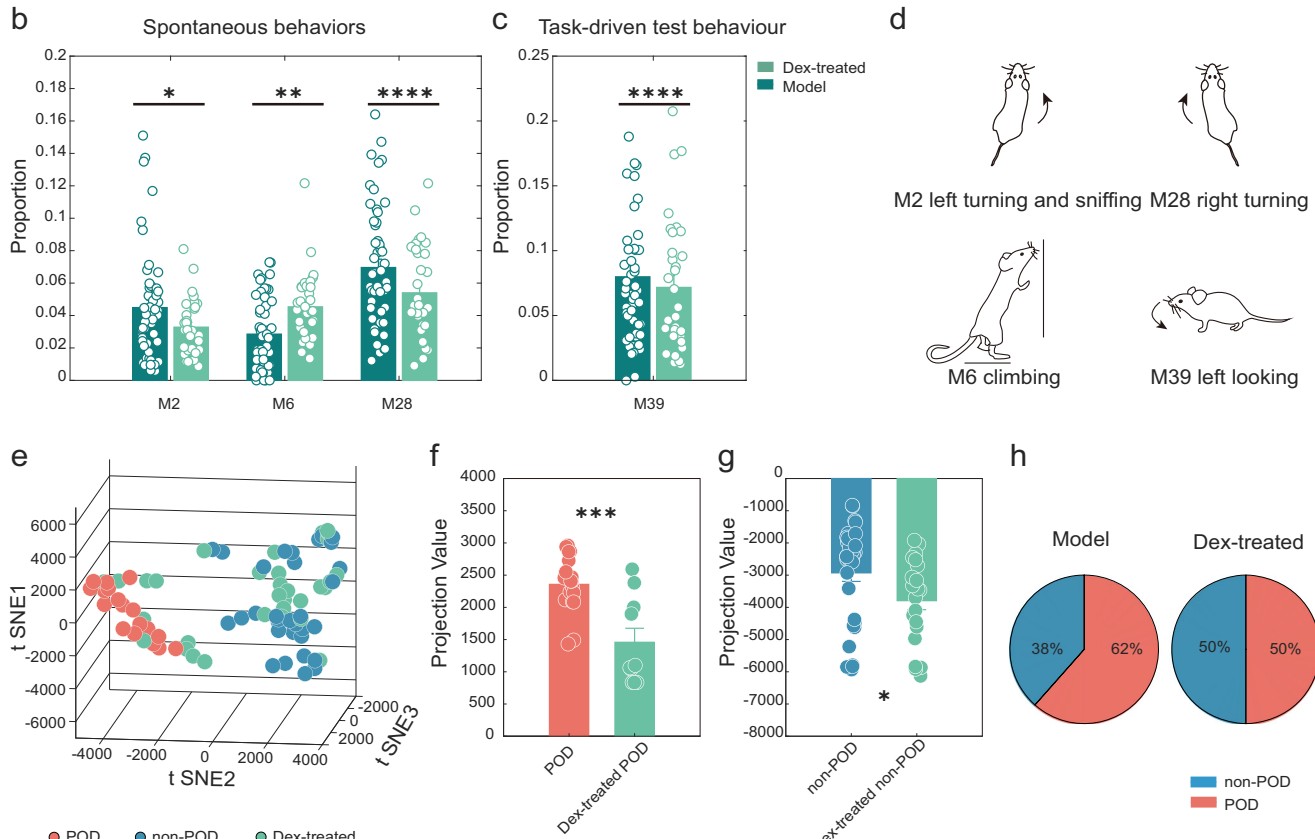

**Fig. 5 Delirium-preventing effects of Dex assessed using machine learning. a** The outline of the design for the Dex-treated test. **b** Behaviours expressed by the Model group and Dex-treated group during the OFT of group monitoring; 40 behavioural traits were computed for each individual. The bold traces and shadows indicate the mean ± SEM (dark green, Model, $n = 52$; light green, Dex-treated, $n = 32$). Three movements showed significant differences between the two groups, and the fractions of the M6 movements that the Dex-treated group mice prefer include M2 (Dex-treated = 0.030 ± 0.003, Model = 0.045 ± 0.005), M6 (Dex-treated = 0.046 ± 0.005, Model = 0.028 ± 0.003), and M28 (Dex-treated = 0.047 ± 0.006, Model = 0.070 ± 0.005). Statistics: two-way ANOVA followed by the Sidak post hoc multiple comparison test, *M2, $P = 0.0215$; **M6, $P = 0.0019$; ****M28, $P < 0.0001$. **c** Behaviours expressed by the Model group and Dex-treated group during the NOR test of group monitoring; 40 behavioural traits were computed for each individual. The bold traces and shadows indicate the mean ± SEM (dark green, Model, $n = 52$; light green, Dex-treated, $n = 32$). One movement showed a significant difference between the two groups (M39; Dex-treated = 0.049 ± 0.006, Model = 0.080 ± 0.008). Statistics: two-way ANOVA followed by the Sidak post hoc multiple comparison test; ****M39, $P < 0.0001$. **d** Schematic diagram of the different behaviours. **e** Low-dimensional representation of the three animal groups (blue, non-POD, $n = 33$; pink, POD, $n = 19$, green, Dex-treated, $n = 32$). The 84 samples in 3D space were dimensionally reduced from 14-dimensional movement fractions, and they are well separated. **f** The projection values were calculated by projecting the points onto the decision boundary (pink, POD, $n = 19$, green, Dex-treated POD, $n = 10$) (POD = 2357 ± 98.48, Dex-treated POD = 1458 ± 217.3; ***$P = 0.0002$, unpaired $t$-test). **g** The projection values were calculated by projecting the points onto the decision boundary (blue, non-POD, $n = 33$, green, Dex-treated non-POD, $n = 22$) (non-POD = −2928 ± 266.6, Dex-treated non-POD = −3792 ± 280.2; *$P = 0.0349$, unpaired $t$-test). **h** Incidence of the Model group and Dex-treated group. All data were presented as mean ± SEM.

demonstrates that the strategy can accurately describe the hierarchical dynamics of delirium-like behaviour in detail and detect delirium-like mice and their subtypes in animal disease models. In addition, using a description of the effects of Dex on postoperative spontaneous behaviours in mice, our framework aims to potentially facilitate clinical medication guidance of Dex in the preclinical stage.

Previous studies reported that anxiety-related performance[13,53], decreased attention[54,55], and cognitive level changes were detected in the POD model. These results are consistent with those of the present study. However, their evidence has shown inconsistencies in many fields. First, the control group in a previous experiment was a "blank" control[13]; thus, the influence on the behaviour of a series of other variables, such as surgical pain and intraoperative hypoxia,

cannot be excluded. This is despite the fact that pain has been implicated as a precipitating factor for POD in past reports[1,5,56]. However, it is undeniable that in classical behavioural tests, the activity-dependent parameters (centre time, mean velocity, etc.) are affected by postoperative pain. Here, we used a recently developed multi-angle behaviour recognition system that showed mice who had undergone the same surgical procedure and anaesthesia displayed different spontaneous behavioural patterns. Applying the Composite Z scores approach, we also detected differences between the model group and the control group. However, this difference disappeared once the POD and non-POD groups were compared at the composite Z-score level. The possible cause of this discrepancy could be that the mice who underwent the same surgical procedure and anaesthesia also displayed different behavioural patterns, which requires a more sensitive tool to detect subtle behavioural changes in delirium-like levels. The spontaneous behaviour, which is of great significance for the diagnosis of the disease models, might be the reason for these phenomena[24]. In our study, we found 14 behaviours with significant differences. By manually reviewing them, we annotated M4 and M10 (Supplementary Movie 1 and Supplementary Movie 2) showing head shaking, which indicates attention disorder[57] and hallucination[58] in previous reports. M8 and M34, associated with walking (Supplementary Movie 3 and Supplementary Movie 4), are considered motor behaviours unrelated to the social exploratory behaviour[59–62]. M9 (Supplementary Movie 5), indicating turning left, and M22, (Supplementary Movie 6) indicating freezing, are related to being in a state of anxiety, fear, and defensiveness[36,63–65]. In addition, M37 indicates gait-change, which is a powerful neurodegenerative disease measurement tool to identify markers of early pathology[66,67] (Supplementary Movie 7). Next, a significant reduction was observed in exploration-related behaviours[60,68,69], annotated as M3 and M36 (rear with exploration), M35 (sniffing) (Supplementary Movie 8, Supplementary Movie 9 and Supplementary Movie 10), and M40 (right-looking) (Supplementary Movie 11). Likewise, M25 and M38 (Supplementary Movie 12 and Supplementary Movie 13), indicating hunching, were significantly reduced in relation to autism[24] and visceral pain[70,71]. M16 (launching) appears as a sudden acceleration, which may be related to the change in the kinematic state of the POD (Supplementary Movie 14).

Our study demonstrates the presence of POD subtypes in mouse models of POD. These results are consistent with those of some studies showing that biperiden-treated rats exhibited two types of alternating behavioural changes, hyperactive and hypoactive states[72]. Several studies have indicated that this manifestation can be elicited in zebrafish[73]. In addition to evidence from animal studies, several clinical trials have reported that for postoperative central nervous system complications after anaesthesia and surgery, there are two phenotypes on the opposite psychomotor side[47,74]. These results support our findings, which reveal that anaesthesia and surgical factors can elicit two clinical subtypes in animal models with different kinematic parameters. These results imply that the changes in the kinematic parameters in the POD subtype are credible.

Finally, previous clinical studies reported that Dex improves delirium after noncardiac surgery[38–41,75]. Limited by quantity and quality, clinical trials investigating the utility of Dex are controversial. Evidence from animal experiments shows that Dex attenuates sepsis-related inflammation and encephalopathy through central $\alpha_{2A}$ adrenergic receptors[76] and may also improve postoperative cognition by exerting anti-inflammatory effects[76,77]. The dose range of Dex is very wide in the study of delirium[17,42,78]. Given that our experimental objective was to validate the efficacy of Dex for non-cardiac surgery, we selected a dose that has been widely reported to have neuroprotective and anti-inflammatory effects[17,51,52]. The dexmedetomidine dose

dependence of the exploration experiment showed that the addition of 25 µg kg$^{-1}$ dexmedetomidine provided the most potent protection that was significantly better than 1 or 10 µg kg$^{-1}$ dexmedetomidine in each brain area[78]. Our results are consistent with those of previous studies showing the involvement of Dex in improving anxiety and cognition-related postoperative behaviours. Currently, the optimal treatment for delirium during critical illness is to avoid the occurrence of risk factors. Thus, our results may expand on the benefits of the use of agents such as Dex[79].

The dynamics of complex symptoms generally include high-dimensional, nonstationary, and nonlinear behaviours, all of which pose fundamental challenges to quantitative understanding. Traditionally, the detection of delirium in animal models is based on the population level[80]. However, the incidence of delirium varies widely. In contrast, the two different subtypes are individual-based measures of disease symptoms and are obscured by averaging response probabilities across trials and individuals. Most importantly, when using the same surgical protocol, the postoperative behavioural responses of mice are different. While many different behavioural tests, such as the Y-Maze, EPM and others, have been defined, the individual response collapses to a binary outcome: a given group either does or does not respond to POD. It is obviously inappropriate to use this rough index to describe a syndrome with complex symptoms like POD. Although OFT and NOR are classical behavioural tests used to detect anxiety and cognition, we used these two scenarios for spontaneous and task-driven behaviours. Unlike centre time, mean speed and total distance, they can be directly caused by neuronal activity[46,81]. Importantly, these two scenarios are widely used to assess spontaneous and task-driven behaviours;[24,36,46,82,83] therefore, we creatively applied these two scenarios to detect the spontaneous and task-driven behaviours of POD.

The present study has several strengths. First, it simulates a condition present in patients, namely, how to behave in situations of an uncertainty subtype. In addition, a description of disease status at different time points in the same mouse was constructed to examine the accuracy of the identification and ensure that unsupervised clustering could reliably detect and classify objects. These behavioural variations provide the backdrop upon which the brain operates, and understanding them is essential for making progress in revealing the neural mechanisms underlying the decline in behavioural and cognitive functions.

Although this study aimed to report the available up-to-date animal behaviour-related evidence, the limitations of this study include not enough studies on the meaning of spontaneous behaviour, which can be attributed to the limited animal behavioural repositories. This can be addressed through the continuous carrying out of animal behaviour studies. In 1974, Teasdale and Jennett's Glasgow coma scale (GCS) was published in *The Lancet*. This standardized bedside tool to quantify consciousness became a medical classic[84]. Altered consciousness is an important clinical symptom of POD, and the GCS score has also become an important scale for the clinical diagnosis of POD[85]. However, quantifying consciousness in animal models is still an important challenge[86]. We used animal sex protocols from other basic studies of POD and selected female mice as subjects[8,13,54,87].

A multi-scaled clustering analysis framework will allow us to expand the scope of our research questions by considering the interactions between different representative behaviours and delirium-like neural mechanisms. Computing mnemonic links may provide an important mechanism to build a cognitive map that stretches beyond direct experience, thus supporting flexible behaviour. The application of a multilayer data analysis strategy offers a valuable opportunity to gain new insights into POD by integrating different interactions and relationships within the

same analytical framework. The use of a multi-layered approach will provide an important new perspective on the complex behaviour that is inherent in POD.

In conclusion, based on the findings of this study, we hypothesize that, in humans, delirium states can be assessed by spontaneous behaviour and can be accurately diagnosed in a timely manner by categorizing subtle states, including facial micro-expressions and body movements. If this hypothesis turns out to be correct, we speculate that a novel diagnostic approach that specifically affects the accuracy and promptness of POD should be developed in the future. The diagnostic method of dependency scale evaluation could then be destabilized, allowing machines to carry out these tasks. This will facilitate risk factor mitigation, identification of potential methods for interventional studies, and informed patient and family counselling.

## Methods

**Animals and experimental groups**. Behavioural tests were performed on 12-month-old C57BL/6J female mice. The mice were maintained under standard animal facility conditions at a temperature of 22–25 °C and a relative humidity of $50 \pm 15\%$. Mice were housed under a reverse 12-h day/12-h night cycle with *ad libitum* access to rodent food. The experimental group was divided into a model group and a control group (Fig. 1a). The model group underwent anaesthesia and exploratory laparotomy. In the control group, neither anaesthesia nor surgery was performed. The POD group came from the model group but had a different behaviour pattern compared with the control group. The non-POD group also came from the model group but had similar behaviour patterns to the control group. The control group was exposed to the same transparent acrylic room as the experimental group and was exposed to air for two hours. The animal-related protocol was performed and approved by the Standing Committee on Animals at Tongji University (approval number: TJBH08423102), and C57BL/6 mice were maintained according to the guidelines in the laboratory animal centre at ShanghaiTech University.

**Animal surgery**. The exploratory laparotomy was used as the model to induce POD[11,13,14]. Briefly, anaesthesia was induced and maintained using $2 \pm 0.2\%$ isoflurane in a transparent acrylic chamber with 90% oxygen. An incision was made along the linea alba. The intestines were exteriorized and the superior mesenteric artery (SMA) was dissected and clamped for 15 s. The clamping was repeated three times. The total exploratory surgery duration was 10 min. After exploration, sterile 4–0 chromic gut sutures were used to suture the peritoneal lining and skin. After the operation, the mice were moved into a transparent acrylic chamber and received $2 \pm 0.2\%$ isoflurane over the 2 h exposure. After the anaesthesia, compound lidocaine cream (2.5% lidocaine and 2.5% prilocaine) was applied to the wound for pain relief, and the next application was 3 h after the operation and 10 min before the start of the behaviour tests. The mice were returned to their home cage upon demonstrating the ability to walk.

**Behaviour tasks**. OFT was used to assess the spontaneous behaviour of mice. NOR was used to assess task-driven behaviour. At the end of each task, the box was cleaned with 75% ethanol before use, and the behavioural test was initiated after ensuring that the ethanol was completely volatilized. The surgery was performed on the seventh day (Supplement Fig. 1). We captured the delirium-like behavioural changes in multi-scale patterns at 6, 30, 54, and 78 h after surgery using a multi-view video capture system[24] (Fig. 1b). During the entire process, a BeA instrument was used for video recording. The resolution of the video is 640*360. The frame of the video is 30.

**Open field test**. The open field test was performed after surgery. As described previously[24], the mice were gently placed in a transparent circular open field with a diameter of 50 cm and a height of 50 cm, and they explored freely for 10 min. Based on the absolute information of the coordinates, we calculated the time that the mice stayed in the centre of the open field, where the radius of the centre of the open field was defined as 50 mm.

**Novel object recognized test**. After the open field test, mice were subjected to the NOR test. As described previously[88], the mice performed exploratory activity in the NOR environment (40 × 40 × 50 cm right quadrangular prism), and two identical objects (training objects) were placed at opposite sides of the box during a training session. The mouse was placed in the middle of the box and allowed to explore for 10 min. During the test, one of the training objects was randomly replaced with an object of a different colour, material, and shape but of the same size and the same difficulty for mice to explore. The mouse was allowed to explore for 10 min. The ratio of time spent on the novel object to the total exploration time on both objects

was calculated.

$$\text{Recognition index} = \frac{(\text{Time spent at novel} - \text{Time spent at familiar})}{(\text{Time spent at novel} + \text{Time spent at familiar})} \quad (1)$$

**Collecting mouse behaviour data**. All processes are described in Fig. 1c, and we reconstructed the 3D skeleton information of the mice using DLC[89], which included 16 body parts of the mice, including the nose, left ear, right ear, neck, left front limb, right front limb, left hind limb, right hind limb, left front claw, right front claw, left hind claw, right hind claw, back, root tail, middle tail, and tail tip. We obtained the 3D skeletons of mice and 40 behavioural skeletons based on BeA[24]. With this 3D skeleton information, we obtained behavioural descriptions of every mouse.

**Extraction and summary of poses data**. Pre-processed behavioural recordings of mice in the OFT were further summarized into descriptions of the pose. A variety of summaries were constructed, based on the parameters described below.

Length: the three-dimensional distance from the mouse's nose to the root of the mouse's tail.

Height: the height of the mouse's neck and the mouse's back.

Front width: the three-dimensional distance from the left forelimb to the right forelimb of the mouse.

Hind width: the three-dimensional distance from the left hindlimb to the right hindlimb of the mouse.

Back speed: the three-dimensional distance of the mouse's back moving each frame rate of change.

Nose speed: the rate of change of the mouse's nose moving a three-dimensional distance per frame.

Left front claw speed: the rate of change of the mouse's left front paw's three-dimensional distance per frame.

Right front claw speed: the rate of change of the mouse's right front paw's three-dimensional distance per frame.

Acceleration: the rate of change of the mouse's back speed.

Angular velocity: the rate of change of the deflection per frame of the direction vector formed by the root of the mouse's tail to the neck.

Spine angle: the included angle formed by the vector pointing from back to neck and the vector pointing from back to tail root.

After obtaining 13 poses, histogram analysis (bins = 50) was performed, and the PMF was calculated. According to the knowledge of signalling, 7 eigenvalues including the mean, median, standard deviation, 25% and 75% quantiles, minimum value and maximum value were extracted for these 13 poses PMF curve. Therefore, we obtain 91 eigenvalues. Here, the resampling work was performed using NumPy in Python, and the Kolmogorov–Smirnov test calculation was performed using scipy in Python.

**Construction of the action sequence data**. As described previously, we define exploration as the occurrence of the following two situations[46,83]. (1) The two-dimensional distance between the mouse's nose and the object in the current frame (regardless of the height difference) is less than 60 mm, and the direction vector formed by the mouse's neck to the nose is the same as the vector formed by the mouse's neck to the centre of the object. The direction vector formed by the mouse's nose is opposite, which means that the angle between the two direction vectors mentioned above should be an acute angle. (2) The two-dimensional distance from the mouse's nose to the object is less than 30 mm, and the height of the mouse's back is greater than the height of the top of the object, which means that the mouse is climbing on the object. Previous reports have suggested that mice showed two states of investigatory behaviour[46,90]: (1) shallow investigation: approach and sniff without any other interactions and (2) deep investigation: approach and sniff with further interactions. For every investigation test, $T_{\text{deep}}$ and $T_{\text{shallow}}$ were calculated as the sum of the durations of all deep investigation and shallow investigation sequences, respectively. We introduced the *DSP* using the relative time a mouse carries out deep investigation compared with the shallow investigation.

$$DSP = \sin^{-1}\left(\frac{T_{\text{deep}} - T_{\text{shallow}}}{T_{\text{deep}} + T_{\text{shallow}}}\right) \quad (2)$$

This index serves as a ratio between time spent in deep investigation and shallow investigation, expressed as an angle. The *DSP* ranges from $-\pi/2$ to $\pi/2$ rad, with $-\pi/2$ rad meaning the mouse exclusively displayed shallow investigation, $\pi/2$ rad meaning that the mouse exclusively displayed deep investigation, and 0 rad meaning equal preference for deep and shallow investigation.

**Composite Z score**. The Composite $Z_{\text{score}}$ was obtained by adding the $Z_{\text{score}}$ of the centre time, total exploration time, and the $Z_{\text{score}}$ of the exploration index. The central area is defined as a circular area with a radius of 10 cm. In addition, the recognition index of mouse exploration was also investigated. It refers to the percentage of time the mouse spent exploring novel objects as a percentage of the

total exploration time.

$$Z_{score} = \frac{Model - MEAN(Control)}{SD(Control)} \quad (3)$$

**Kolmogorov–Smirnov test**. According to the PMF, the empirical distribution function (EDF) was obtained. Based on the EDF, random samplings were performed to test the distribution differences. We resampled 500 samples from the EDF, and a two-sided Kolmogorov-Smirnov test was performed to compare the EDF differences between the two groups. Detailed operations are as follows.

We denote the real distribution of one group as $F(x)$ and the other as $G(x)$. We took 500 samples $x_1, x_2, ..., x_{500}$ from one group through the EDF of F(x), and $y_1, y_2, ..., y_{500}$ from another group through the EDF of G(x). The null hypothesis $H_0$ is $F(x) = G(x)$, and the alternative hypothesis $H_1$ is $F(x) \neq G(x)$. We merged and sorted the sampled data to depict respective EDF $F_0(x)$ and $G_0(x)$ and found the K-S statistic $D$ through finding the maximum value by subtracting the two EDFs, i.e., $D = \max |F_0(x) - G_0(x)|$. After obtaining $D$, we decided whether to reject the $H_0$ by looking up the statisitical table of $D_{n,\alpha}$, where $n = 1000$ represents the number of samples and $\alpha$ is set to 0.05. Notably, $F_0(x)$ can be obtained by a sorted sample $x_{(1)}, x_{(2)}, ..., x_{(500)}$ and

$$F_0(x) = \begin{cases} 0, & \text{for } x < x_{(1)} \\ \frac{k}{n}, & \text{for } x_{(k)} \leq x < x_{(k+1)}, k = 1, 2, .., 500 \\ 1, & \text{for } x \geq x_{(500)} \end{cases} \quad (4)$$

and $G_0(x)$ can be obtained similarly. The whole process was completed by Python.

**Delirium-like behaviour space**. The proportion of 14 movements $x_{i_1}, x_{i_2}, ..., x_{i_{14}}$ of a certain POD mouse $i$ (Fig. 1b) constitutes a vector $\mathbf{X}_i$ in 14D delirium-like behavioural vector space, i.e. $\mathbf{X}_i = [x_{i_1}, x_{i_2}, ..., x_{i_{14}}]^T, i = 1, 2, ..., n$, where $n$ is the total number of POD mice. $\mathbf{X} = [X_1, X_2, ..., X_n]$ is a $14 * n$ matrix representing the propotion of all POD mice movements. We then used t-SNE to reduce the feature dimensions of X from 14 to 3 dimensions, as follows:

$$Y = f_{t-SNE}(x) \quad (5)$$

where $Y$ represent a $3 * n$ matrix generated by the t-SNE algorithm. The $f_{t-SNE}()$ includes the parameters n_neighbors set to 6, which are robust enough to change across a wide range and discriminate between POD and non-POD mice in the delirium-like behavioural space. To quantify the group differences, we fitted a classification model to $Y$ using a linear kernel by the svm.SVC function in Python with default parameters.

**Effect prediction based on machine learning**. Classification based on behavioural summaries was performed using logistic regression as implemented in the ClassificationLeaner Matlab package. To avoid overfitting, we performed a 5-fold cross-validation and then performed 30 iterations until the minimum classification error converged. To evaluate performance, confusion matrices and a minimum classification error curve were computed (Supplement Fig. 5).

**Statistics and reproducibility**. All statistical analyses were performed using MATLAB (R2020b, MathWorks, Natick, MA, United States of America), GraphPad Prism 8, and Python (v3.8.5; Python Software Foundation). The movement fractions data were normally distributed, with homogeneous variances; thus, a two-way ANOVA followed by the Sidak post hoc multiple comparison tests was used to compare the differences among the groups. Data were considered statistically significant when $P$ values were less than 0.05. The asterisks denote statistical significance: $*P < 0.05$, $**P < 0.01$, and $***P < 0.001$, $****P < 0.0001$. Unless stated otherwise, values are presented as the mean ± SEM.

**Reporting summary**. Further information on research design is available in the Nature Portfolio Reporting Summary linked to this article.

## Data availability
3D skeleton trajectories associated with spontaneous behaviour tests are available in the figshare repository[91] (https://doi.org/10.6084/m9.figshare.23538255). The samples of supplementary videos are available in the figshare repository[92] (https://doi.org/10.6084/m9.figshare.23641914). All data related to this study have been included in the article and its supplementary information. Source data used to generate bar figures are available as Supplementary Data 1.

## Code availability
The relevant data are available upon reasonable request.

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

## Acknowledgements

We acknowledge the Natural Science Foundation of China and Natural Science Foundation of Shanghai and Shanghai Fourth People's Hospital, School of Medicine, Tongji University for providing funding support for the current work. This work was supported by the National Natural Science Foundation of China [grant number: 82271223], Shanghai Municipal Committee of Science and Technology [grant number: 23XD1422900] and the Shanghai Fourth People's Hospital, School of Medicine, Tongji University [grant numbers: sykyqd01902, SY-XKZT-2021-2001].

## Author contributions

S.C., Y.W. and Z.G. collected the data and drafted the manuscript, and all authors take responsibility for the integrity of the data and accuracy of the data analysis. J.T. plotted the figures. L.X. guided the writing of the original draft. J.H. and C.L. conceptualized and designed the study. All authors have reviewed and approved the final manuscript.

## Competing interests

The authors declare no competing interests.
