## [Peer Review File · Communications Biology]

Reviewers' comments:

Reviewer #1 (Remarks to the Author):

The paper by Cao et al. introduces a novel method for behavior quantification in the animal model of postoperative delirium POD.

This study capitalizes on previous findings by Huang et al. that established a 3d hierarchical framework in animal behavior. The authors propose to evaluate task-induced and spontaneous behavioral activities. This combination, without a doubt, gives a more in-depth behavioral assessment as compared to just a task-driven behavioral assessment. As a result, the authors were able to discern behavioral differences otherwise not detectable by conventional methods.

The authors describe in detail the definition of various behavioral components and nomenclature.

Deep lab cut, an open-source platform with 16 body parts definitions, was used for behavior definitions.

The multidimensional data space was reduced using t-SNE.

One of the hurdles of the preclinical assessments of various drug candidates is demonstrating efficacy through improved behavioral outcomes. Quite often, such assessments fail to note significant behavioral differences due to the high variability and low sensitivity of task-oriented tests. Using the reported methodology, the authors can pull apart behavioral differences in POD vs Non-POD animals after Dex treatment.

Comments

It's not clear the length of the video recordings. It appears that each of the observations didn't last longer than 10 min with a 30 fps time resolution. The resolution of the video was not specified.

There is no clear description of the n number in animal experiments and machine learning. It varies from 11 to 52. But In machine learning experiments, it goes beyond 1000. In some cases, n is defined as the number of animals in others by the number of observations of behavioral activities. One of the options is to make a table showing how many animals were used for each experimental group. For machine learning, 300-fold cross-validation was used. What were the data frame dimensions? A flow chart of data preprocessing will be informative in this regard.

In my opinion, a table defining the n for each experiment will help the reader to understand the methodology better.

In figures 2c, 3b, 3s b&c, and 5s a&b, the figure legends do not provide enough information. There is some much of an overlap that, in my opinion, different means of showing these data are needed.

In figure 5s, the confusion matrix data suggest perfect model performance. At the same time, in the same figure panel d some mice converted from sick to healthy and vice versa at different points after surgery, suggesting mistakes in model definitions.

In the discussion section, the authors state: "... limitations of this study include the failure of the authors to cite enough literature on some behaviors...". If the authors think that more literature needs to be cited, what stopped them from doing so?

Reviewer #2 (Remarks to the Author):

In the manuscript the authors based on recent advanced animal behavior recording and analyzing techniques to present a multi-scaled behavioral phenotyping framework for disease animal model assessment. The authors used several aspects of behavioral feature to demonstrate the benefits of comprehensive behavioral profiling method for the identification of animal diseases, classification of disease subtypes, and evaluation of drug efficacy etc. Most significantly, the quantifiable behavioral data presented in this study are consistent with symptoms of the disease in other species, especially humans.

This experiments of this work was well designed, the data analysis and visualization were novel, and

the manuscript was well organized. I believe this work will inspire more researchers in this field to characterize the behaviors of animal disease models across-species. Before this manuscript could be accepted for publication in *Communications Biology*, there are minor flaws that need to be improved. Here below are a few comments that I hope will help to improve this work.

1) The novelty of this study is to emphasize the multi-scale, hierarchical and dynamic behavioral properties, which can better represent the disease symptoms of animals compared with the traditional static behavioral parameters. Therefore, in the second paragraph, the authors introduced that traditional behavioral assessment methods mainly use the behavior tests such as EPM, OFT, and NOR, by measuring static parameters, such as the time an animal spends performing a specific task in a confined chamber. Then in the third paragraph, the authors cited many literatures, explaining that animal behavior should be hierarchical, dynamic, time-varying, which has the human language-like characteristics. And explains why animal behavior has this characteristic from the neural-behavioral correlation relationship aspect.

My suggestion for the third paragraph is that there needs a clearer explanation of why multi-scale, hierarchical, dynamic behavioral features are better for evaluating POD. It needs to be elaborated from the relationship between neural activity, symptoms, and behavior. I recommend first explaining the relationship between disease symptoms and abnormal neural activity, then demonstrating that symptoms are reflected in multi-aspects behavioral characteristics, and finally emphasizing that animal behavior is shown to be such a multi-level structure, thus spontaneous behavior, task-driven behavior is necessary for POD assessment.

2) Line 286: "we labeled the 3D skeleton information of the mice using DeepLabCut (DLC) which included 16 body parts of the mice with 3D coordinates".

I think here should be label 2D pose for four single views, not 3D skeleton. The 3D coordinates are reconstructed from four 2D coordinates. Therefore, the process of this part should be rewrite and add these details. Accordingly, in Fig. 1c, the other 3 views of the DLC pose estimation images need to be added.

3) Line 431: 11 postural features do not seem to correspond to Fig. S4a-m, 13 postural features? Besides, "eigenvalue" should be replaced with "features". Because eigenvalue is a term for linear algebra, not for certain behavioral parameters of an animal.

4) Both spellings of "behaviour" and "behavior" exist in manuscripts, the spelling needs to be improved.

Reviewer #3 (Remarks to the Author):

Although the authors present a lot of data and I think their motivation for characterizing behaviors in rodents post-anesthesia might be good, there are too many things wrong with this manuscript to describe in detail. This review hits the highlights, however even if these issues were remediated the other glaring errors might become even more apparent.

First, the logical flow of the paper is in question. As best I can tell the authors intend to use AI and video data from multiple cameras to demonstrate efficacy of dexmedetomidine in prevention or treating delirium in their rodent model. This is "the cart before the horse". First, there is no standard rodent model for post-operative delirium. I think the authors are attempting to propose that their methodology can detect a rodent version of post-operative delirium, but it is not explicitly stated like that. The text of the Results section cannot be followed by the reader. The authors state that "delirium-like behaviors" (not described at all) are able to be detected with their methodology and seven separate behaviors obtained from their methods (these seven not well described) demonstrate rodent delirium. In order to legitimately make the claim that these behavioral tests that "diagnose"

delirium in mice the authors would have to show more commonality with human delirium. For example are these abnormal recoveries more likely in aged animals or long surgeries? The figure doesn't shed light on the situation, why does left turning predict delirium and not right-turning? What does specifically looking to the right have to do with the mechanism of delirium? The videos reference are so short (some less than 1 second)– there is no context for any of these behaviors. I have read the results section several times but I cannot understand how the authors progress from these associations to a "model" group and a red/green group which is some sort of group intended to represent a cohort with an expected delirium rate? Maybe. The description is hard to follow. Additionally, the fact that dexmedetomidine decreases these "abnormal" behaviors does not "prove" these behaviors are delirium. Although delirium rates might be lowered with dexmedetomidine it is possible that in this model, dexmedetomidine is causing an overall increase in quiescence. Overall activity level is not really controlled for in the behaviors indicative of rodent delirium.

Second, in multiple places throughout the paper the language is very confusing and non-standard for rodent researchers in the area. The concepts of "model group" and "control group" and red/green designations are not clear at all. Additionally, I am not sure what the authors mean when they write "static parameters" for behavior? While some of this may be due to translation services being used (were translation services used?). It sounds a little like the kind of word salad that come from AI technology. Most of the key points were wordy and awkwardly described. Reviewing the text of this manuscript was arduous. Obvious examples of this are in the abstract where the authors describe delirium as "pernicious". While, delirium is often underdiagnosed and can lead to an escalation of care, it is (itself) not progressive per se but rather associated with the development of cognitive decline in the long-term, such as an exacerbation Alzheimer's Disease and related dementias. The next sentence of the abstract is arguably even harder to understand. "Traditional behavior analysis mostly focuses on static parameters, which lost the structure of animal behavior, or is based on limited observations of the spatial positions of the mice and their derivatives." It has awkward syntax, a verb conjugation error, and confusing. I think the authors are trying to say that: Most rodent behavioral analysis programs focus on identifying particular states of behavior that occur intermittently (i.e., grooming or freezing) and does not incorporate temporal dynamics or multiple camera angles. Honestly, I am not really sure what they are attempting to convey, but I provide my interpretation as to their meaning based on how I would motivate this project if it were my data.

In summary, there may be some merit in the data – but I cannot understand the scientific narrative nor the methods to give this manuscript a proper ranking. Nor can I be very specific with what needs to be changed, to improve the chances of acceptance.

Reviewer #1 (Remarks to the Author):

The paper by Cao et al. introduces a novel method for behavior quantification in the animal model of postoperative delirium POD.

This study capitalizes on previous findings by Huang et al. that established a 3d hierarchical framework in animal behavior. The authors propose to evaluate task-induced and spontaneous behavioral activities. This combination, without a doubt, gives a more in-depth behavioral assessment as compared to just a task-driven behavioral assessment. As a result, the authors were able to discern behavioral differences otherwise not detectable by conventional methods.

The authors describe in detail the definition of various behavioral components and nomenclature.

Deep lab cut, an open-source platform with 16 body parts definitions, was used for behavior definitions.

The multidimensional data space was reduced using t-SNE.

One of the hurdles of the preclinical assessments of various drug candidates is demonstrating efficacy through improved behavioral outcomes. Quite often, such assessments fail to note significant behavioral differences due to the high variability and low sensitivity of task-oriented tests. Using the reported methodology, the authors can pull apart behavioral differences in POD vs Non-POD animals after Dex treatment.

A: Thank you for the positive comments and constructive suggestions to improve our paper.

Comments:

Q1: It's not clear the length of the video recordings. It appears that each of the observations didn't last longer than 10 min with a 30fps time resolution. The resolution of the video was not specified.

A1: We are sorry for the unclear description of the video recording. We recorded 10 mins videos of mice during OFT and NOR with 30fps in 640*360 resolution. The supplementary videos showed 40 video clips of decomposition behavior. Due to the file

size limitation of the submission system, we could not upload the original video, so we compressed the clarity (150MB to 5MB) to ensure a smooth upload (Supplement video21-24). They are the sample videos from four perspectives of the same experiment, and we labeled the movement number to each frame (Supplement Table 2).

According to your suggestion, we will add detail video recording information in the revised manuscript (Line139-140).

Q2: There is no clear description of the n number in animal experiments and machine learning. It varies from 11 to 52. But In machine learning experiments, it goes beyond 1000.

A2: We apologize for the unclear description of n number in animal experiments and machine learning. We add a table (Supplement Table1) with detail n number in animal experiments and machine learning. Due to the long time period of animal experiments, it is difficult to get a sample size of hundreds. And the current classification results show that the two clusters are well separated with very low classification error, in which additional sample size won't significantly decrease the error. Therefore, we think this amount of data is sufficient for a classification problem.

Q3: In some cases, n is defined as the number of animals in others by the number of observations of behavioral activities. One of the options is to make a table showing how many animals were used for each experimental group.

A3: We are sorry for the unclear description of n number. In our research, every mouse will experience OFT and NOR 4 times within 78h as follow.

a

As your suggestion, we provide a table with sample sizes of each experiment in the supplementary figures section.

Supplement table1. The sample size for each experiment.

Model group(n=52)	POD group(n=19)	Hyperactive subgroup(n=8)
		Hypoactive subgroup(n=11)
	Non-POD group(n=33)	

Q4: For machine learning, 300-fold cross-validation was used. What were the data frame dimensions? A flow chart of data preprocessing will be informative in this regard.

A4: Sorry for this erratum “300-fold cross-validation was used”. We tried to explain the Figure S5 in detail, in fact, we used 5-fold cross-validation, and then performed 30 iterations until the minimum classification error converged. This error has been corrected in the revised manuscript (Line248-250). The data frame dimension is 14. We really appreciate your comments, and we have provided a flow chart of data preprocessing in the supplementary figure as your suggestion. It is shown as follows:

We mixed the sample points of the model group (isoflurane anesthesia and surgery) and the Dex-treated group (Dexmedetomidine and isoflurane anesthesia and surgery) for classification, trying to find the similar trend between them. In the classification

process, through the minimum classification error and confusion matrix, we show that pod and non-POD are well separated, and in the process of low-dimensional space visualization, POD and non-POD are well separated, which reflects the interpretability of this model to a certain extent.

Since we apply the same mapping rule to the Dex-treated group and the model group, which means that our criteria are consistent, it is reasonable to assume that the sample point of the Dex-treated group with a close distance to pod is Dex-treated-POD, and that of the Dex-treated group with a close distance to non-POD is Dex-treated-non-POD.

Q5: In my opinion, a table defining the n for each experiment will help the reader to understand the methodology better.

A5: Thank you for your suggestion. We provide a table with sample sizes of each experiment in the supplementary figures section.

Q6: In figures 2c, 3b, 3s b&c, and 5s a&b, the figure legends do not provide enough information. There is some much of an overlap that, in my opinion, different means of showing these data are needed.

A6: Thank you for your suggestion. We have removed the duplicated parts according to your suggestion. We have made changes to the figures and legends based on your comments.

Q7: In figure 5s, the confusion matrix data suggest perfect model performance. At the same time, in the same figure panel d some mice converted from sick to healthy and vice versa at different points after surgery, suggesting mistakes in model definitions.

A7: We are sorry for the inappropriate expression of this part that make you confused to our result. According to your comments, we revised the Supplement figure5 in the revised manuscript. As follow:

In addition, we explain your confusion as follow:

First, the training set of confusion matrix was the 14 significant different movements from POD group and non-POD group. And the test set is these 14 movements of the 8 Dex-treated mice in four days. The detection was repeated four times in four days for each mouse, in other words, the sample size of test set is 32. We performed four days because the onset time and recovery time are not uniform, which is an important feature of POD.

In supplement figure5, panel c is showing the state of test set in four days (POD or non-POD), there are some mice from sick to healthy and vice versa at different points after surgery. The confusion matrix is derived from the cross-validation results. There is no conflict with the classification error of the training set shown by the confusion matrix. To avoid confusing the reader, we have modified the figure in the revised manuscript.

Q8: In the discussion section, the authors state: "... limitations of this study include the failure of the authors to cite enough literature on some behaviors...". If the authors think that more literature needs to be cited, what stopped them from doing so?

A8: We are sorry for the wrong expression of this sentence. We have rewritten the sentence "the limitations of this study include there are not enough literatures about the meaning of spontaneous behaviour, which can be attributed to the limited animal behavioural repositories. This can be addressed through the continuous carrying out of animal behaviour studies" (Line 579-581). In addition, we explain your confusion as follow:

In our manuscript we find a lot of behaviors in POD mice that are statistically different from non-POD mice. Base on the observations to video clips of movements, we name them, e.g., rear with exploration. However, in the pursuit of the biological significance of these movements, limited by the relatively few reports on the behaviour of animals, we can only interpret the behavioral significance of these different movements through these existing reports.

Reviewer #2 (Remarks to the Author):

In the manuscript the authors based on recent advanced animal behavior recording and analyzing techniques to present a multi-scaled behavioral phenotyping framework for disease animal model assessment. The authors used several aspects of behavioral feature to demonstrate the benefits of comprehensive behavioral profiling method for the identification of animal diseases, classification of disease subtypes, and evaluation of drug efficacy etc. Most significantly, the quantifiable behavioral data presented in this study are consistent with symptoms of the disease in other species, especially humans.

This experiments of this work was well designed, the data analysis and visualization were novel, and the manuscript was well organized. I believe this work will inspire more researchers in this fields to characterize the behaviors of animal disease model across-species. Before this manuscript could be accepted for publication in the Communications Biology, there are minor flaws that need to be improved. Here below a few comments that I hope will help to improve this work.

1) The novelty of this study is to emphasis the multi-scale, hierarchical and dynamic behavioral properties, which can better represent the disease symptoms of animals compared with the traditional static behavioral parameters. Therefore, in the second paragraph, the authors introduced that traditional behavioral assessment methods mainly use the behavior tests such as EPM, OFT, and NOR, by measuring static parameters, such as the time an animal spends performing a specific task in a confined chamber. Then In the third paragraph, the authors cited many literatures, explaining that animal behavior should be hierarchical, dynamic, time-varying, which has the human language-like characteristics. And explains why animal behavior has this characteristic from the neural-behavioral correlation relationship aspect.

My suggestion for the third paragraph is that there needs a clearer explanation of why multi-scale, hierarchical, dynamic behavioral features are better for evaluating POD. It

needs to be elaborated from the relationship between neural activity, symptoms, and behavior. I recommend first explaining the relationship between disease symptoms and abnormal neural activity, then demonstrating that symptoms are reflected in multi-aspects behavioral characteristics, and finally emphasizing that animal behavior is shown to be such a multi-level structure, thus spontaneous behavior, task-driven behavior is necessary for POD assessment.

A: Thank Reviewer 2 for the positive comments and constructive suggestions to improve our paper. According to your suggestion, we have changed the expression of this part (Line65-82).

2) Line 286: “we labeled the 3D skeleton information of the mice using DeepLabCut (DLC) which included 16 body parts of the mice with 3D coordinates”.

A: Thank you for your suggestion. According to your suggestion, we have added this part in the revised manuscript (Line 157-163).

I think here should be label 2D pose for four single views, not 3D skeleton. The 3D coordinates are reconstructed from four 2D coordinates. Therefore, the process of this part should be rewrite and add these details. Accordingly, in Fig. 1c, the other 3 views of the DLC pose estimation images need to be added.

A: Thank you for your suggestion. According to your suggestion, we have changed the flow chart.

3) Line 431: 11 postural features do not seem to correspond to Fig. S4a-m, 13 postural features? Besides, “eigenvalue” should be replaced with “features”. Because eigenvalue is a term for linear algebra, not for certain behavioral parameters of an animal.

Thank you for your suggestion. We apologize for the wrong writing and have corrected it in the revised manuscript. Here we used “eigenvalue” to describe the 7 values extract from the PMF. To avoid the misunderstanding, we revised this part in the revised manuscript (Line185-188).

4)Both spellings of “behaviour” and “behavior” exist in manuscripts, the spelling needs to be improved.

Q: We are sorry for our carelessness and we will unify the spelling of the words in the revised manuscript.

Although the authors present a lot of data and I think their motivation for characterizing behaviors in rodents post-anesthesia might be good, there are too many things wrong with this manuscript to describe in detail. This review hits the highlights, however even if these issues were remediated the other glaring errors might become even more apparent.

A: Thank you for your approval of the experimental content and methods. We will try

our best to answer your suggestions.

Q1: First, the logical flow of the paper is in question. As best I can tell the authors intend to use AI and video data from multiple cameras to demonstrate efficacy of dexmedetomidine in prevention or treating delirium in their rodent model. This is “the cart before the horse”.

A1: Thank you for your advice. Firstly, dexmedetomidine is one of the most promising drugs for POD prevention. The dexmedetomidine dose we selected was based on a survey of previous reports. They were proved that dexmedetomidine at 25ug/kg had a good neuroprotective effect. Therefore, in this study, we made a detailed comparison of the effect of dexmedetomidine on postoperative behaviors, including postoperative spontaneous behaviors and cognitive-related task-driven behaviors. The results showed that dexmedetomidine did have a significant effect on some movements. This may be one of the clues in the future to explain the mechanism of dexmedetomidine in preventing delirium. Secondly, the evaluation of the effects of dexmedetomidine was an exploration of the effects of Dex in our study, not an attempt to validate the findings that POD mice produce abnormal behaviors in spontaneous and task-driven behaviors.

Q2: First, there is no standard rodent model for post-operative delirium. I think the authors are attempting to propose that their methodology can detect a rodent version of post-operative delirium, but it is not explicitly stated like that. The text of the Results section cannot be followed by the reader.

A2: Thank you for your suggestion and affirmation. As you mentioned, there is no gold standard for the diagnosis of POD in rodents at present. It is undeniable that POD has been discussed endlessly, and the research on POD mechanism is also an important issue in the field of anesthesia. We found that previous studies used the center time of open-field tests to evaluate POD. Center time uses the habits of rodents as a classic behavioral indicator of anxiety. Whether it could exhibit the complex manifestations of POD attracted our attention.

Since there is no gold standard for animal behavior in POD, we used unsupervised clustering to distinguish between POD and non-POD groups by similarity to mice with normal behavior (control group). And we found significant differences in spontaneous and task-driven behaviour between POD and non-POD groups. In addition, we showed that there were indeed cognitive changes in the POD group with the cognitive assessment method reported in *Cell*[2].

We believe that our detailed evaluation of spontaneous behaviors and task-driven behaviors in animals after surgery is of great significance to the basic research of POD and will be used by other researchers in the future.

Sorry for giving you a bad reading experience, we have rewritten the results section to make it easier to understand in the revised manuscript.

Q3: The authors state that “delirium-like behaviors” (not described at all) are able to be detected with their methodology and seven separate behaviors obtained from their methods (these seven not well described) demonstrate rodent delirium. In order to legitimately make the claim that these behavioral tests that “diagnose” delirium in mice the authors would have to show more commonality with human delirium. For example are these abnormal recoveries more likely in aged animals or long surgeries?

A3: Sorry for the unclear description, we have made a more detail description in the revised manuscript (Line517-529). In our study, to increase the credibility of the experimental results, we also evaluated the traditional behavior of mice, which was consistent with the previously reported behavioral results of mice. And for clinical application, we also found that in our model, the POD mice met the characteristics of CAM scale: early onset, fluctuating state, cognitive decline, psychomotor changes, attention decline, etc. Therefore, we believe that in our study, partial clinical features of POD were reproduced in mice using spontaneous and fine behaviors.

Q4: The figure doesn't shed light on the situation, why does left turning predict delirium and not right-turning? What does specifically looking to the right have to do with the mechanism of delirium?

A4: In our research, only the left turning was statistically different between two groups, not the right turning. Moreover, statistical significance sometimes does not imply biological significance, and it may be that the biological significance of the action needs further discovery and interpretation, or it may not be, as described in limitation (Line579-581). Our future studies will therefore focus on these movements that are both differential and interpretable.

Q5: The videos reference are so short (some less than 1 second)– there is no context for any of these behaviors.

A5: We recorded 10 min video of mice during OFT and NOR with 30fps. The supplementary video showed 40 video clips of decomposition behavior. Due to the file size limitation of the submission system, we could not upload the original video, so we compressed the clarity (150MB to 5MB) to ensure a smooth upload (Supplement video21-24). They are videos from four perspectives of the same experiment, and we labeled the movement number to each frame (Supplement Table 2).

Q6: I have read the results section several times but I cannot understand how the authors progress from these associations to a “model” group and a red/green group which is

some sort of group intended to represent a cohort with an expected delirium rate? Maybe. The description is hard to follow.

A6: Thank you for your efforts in our research. According to your comments, we have changed the figures, legends and descriptions of results in this part. The model group was mice that underwent anesthesia and surgery, and the control group was mice that did not undergo anesthesia and surgery. Red group or green group was the subcluster of the cluster analysis. They were not equal to the model and control groups. In the modified figure, we changed the presentation of this section. As follow:

Q7: Additionally, the fact that dexmedetomidine decreases these “abnormal” behaviors does not “prove” these behaviors are delirium. Although delirium rates might be lowered with dexmedetomidine it is possible that in this model, dexmedetomidine is causing an overall increase in quiescence. Overall activity level is not really controlled for in the behaviours indicative of rodent delirium.

A7: According to your suggestion, we evaluated the effect of dexmedetomidine on postoperative velocity, and the results showed that there was no significant difference between the two groups.

Q8: Second, in multiple places throughout the paper the language is very confusing and non-standard for rodent researchers in the area. The concepts of “model group” and “control group” and red/green designations are not clear at all.

A8: Thank you for your comments. We have made modifications to the naming problem in the revised manuscript, and described the experimental grouping in detail in the methods section (Line110-114). In addition, in order to better display the results, we modified Figure2 in the revised manuscript. We have changed the representation of clustering results in the hope that this can better display our results.

Q9: Additionally, I am not sure what the authors mean when they write “static parameters” for behavior?

A9: Sorry for the unclear description to “static parameters”. Here we try to use static parameters to refer to the traditional behavior of center time, open arm time, total distance and so on. Their common characteristic is that they ignore the inherent dynamics of the behavior itself. In other words, previous studies only looked at how long the mice were in the center, not what they were doing. In the abstract, we do not explain the static parameters in detail due to the word limit. But we talked about it in detail in the introduction (Line 56-64). If this still confuses you, we can change the wording, but in the revised manuscript we still use 'static parameters' to describe the general characteristics of traditional behaviour.

Q10: While some of this may be due to translation services being used (were translation services used?). It sounds a little like the kind of word salad that come from AI technology. Most of the key points were wordy and awkwardly described.

Reviewing the text of this manuscript was arduous. Obvious examples of this are in the abstract where the authors describe delirium as “pernicious”. While, delirium is often underdiagnosed and can lead to an escalation of care, it is (itself) not progressive per se but rather associated with the development of cognitive decline in the long-term, such as an exacerbation Alzheimer’s Disease and related dementias.

A10: We are sorry for bringing you a bad experience during reviewing our manuscript. As you said, this manuscript has been professionally polished. According to your suggestions, we have made modifications in the revised manuscript.

The next sentence of the abstract is arguably even harder to understand. “Traditional behavior analysis mostly focuses on static parameters, which lost the structure of animal behavior, or is based on limited observations of the spatial positions of the mice and their derivatives.” It has awkward syntax, a verb conjugation error, and confusing. I think the authors are trying to say that: Most rodent behavioral analysis programs focus on identifying particular states of behavior that occur intermittently (i.e., grooming or freezing) and does not incorporate temporal dynamics or multiple camera angles.

A: Sorry for making you confused by this sentence. In view of the word limitation in the abstract section, we used 'static parameters' to summarize the previous behavioral

detection methods of POD. We rewrote the sentence in the abstract of revised manuscript (“Traditional behaviour analysis mostly focused on static parameters. They are based on the limited observations of mice’s positions particular states, which lose the structure of behaviours”) (Line24-25). In the introduction, we expand on the characteristics of traditional behaviour (Line 56-64). Static parameters correspond to our elaboration in the next paragraph that behaviour itself is structured and is fluid (Line 68-75).

Q11: Honestly, I am not really sure what they are attempting to convey, but I provide my interpretation as to their meaning based on how I would motivate this project if it were my data.

A11: We sincerely thank you for your comments and suggests to our research. We have tried our best to answer your questions and made modifications according to your suggestions.

1. Nagai, J., et al., *Hyperactivity with Disrupted Attention by Activation of an Astrocyte Synaptogenic Cue*. *Cell*, 2019. **177**(5).
2. Shan, W., et al., *Critical role of UQCRC1 in embryo survival, brain ischemic tolerance and normal cognition in mice*. *Cellular and Molecular Life Sciences : CMLS*, 2019. **76**(7): p. 1381-1396.

REVIEWERS' COMMENTS:

Reviewer #1 (Remarks to the Author):

In my opinion, the authors did a good job addressing the reviewers comments.

Reviewer #2 (Remarks to the Author):

I have no more questions

Reviewer #1 (Remarks to the Author):

The paper by Cao et al. introduces a novel method for behavior quantification in the animal model of postoperative delirium POD.

This study capitalizes on previous findings by Huang et al. that established a 3d hierarchical framework in animal behavior. The authors propose to evaluate task-induced and spontaneous behavioral activities. This combination, without a doubt, gives a more in-depth behavioral assessment as compared to just a task-driven behavioral assessment. As a result, the authors were able to discern behavioral differences otherwise not detectable by conventional methods.

The authors describe in detail the definition of various behavioral components and nomenclature.

Deep lab cut, an open-source platform with 16 body parts definitions, was used for behavior definitions.

The multidimensional data space was reduced using t-SNE.

One of the hurdles of the preclinical assessments of various drug candidates is demonstrating efficacy through improved behavioral outcomes. Quite often, such assessments fail to note significant behavioral differences due to the high variability and low sensitivity of task-oriented tests. Using the reported methodology, the authors can pull apart behavioral differences in POD vs Non-POD animals after Dex treatment.

A: Thank you for the positive comments and constructive suggestions to improve our paper.

Comments:

Q1: It's not clear the length of the video recordings. It appears that each of the observations didn't last longer than 10 min with a 30fps time resolution. The resolution of the video was not specified.

A1: We are sorry for the unclear description of the video recording. We recorded 10 mins videos of mice during OFT and NOR with 30fps in 640*360 resolution. The supplementary videos showed 40 video clips of decomposition behavior. Due to the file

size limitation of the submission system, we could not upload the original video, so we compressed the clarity (150MB to 5MB) to ensure a smooth upload (Supplement video21-24). They are the sample videos from four perspectives of the same experiment, and we labeled the movement number to each frame (Supplement Table 2).

According to your suggestion, we will add detail video recording information in the revised manuscript (Line139-140).

Q2: There is no clear description of the n number in animal experiments and machine learning. It varies from 11 to 52. But In machine learning experiments, it goes beyond 1000.

A2: We apologize for the unclear description of n number in animal experiments and machine learning. We add a table (Supplement Table1) with detail n number in animal experiments and machine learning. Due to the long time period of animal experiments, it is difficult to get a sample size of hundreds. And the current classification results show that the two clusters are well separated with very low classification error, in which additional sample size won't significantly decrease the error. Therefore, we think this amount of data is sufficient for a classification problem.

Q3: In some cases, n is defined as the number of animals in others by the number of observations of behavioral activities. One of the options is to make a table showing how many animals were used for each experimental group.

A3: We are sorry for the unclear description of n number. In our research, every mouse will experience OFT and NOR 4 times within 78h as follow.

a

As your suggestion, we provide a table with sample sizes of each experiment in the supplementary figures section.

Supplement table1. The sample size for each experiment.

Model group(n=52)	POD group(n=19)	Hyperactive subgroup(n=8)
		Hypoactive subgroup(n=11)
	Non-POD group(n=33)	

Q4: For machine learning, 300-fold cross-validation was used. What were the data frame dimensions? A flow chart of data preprocessing will be informative in this regard.

A4: Sorry for this erratum “300-fold cross-validation was used”. We tried to explain the Figure S5 in detail, in fact, we used 5-fold cross-validation, and then performed 30 iterations until the minimum classification error converged. This error has been corrected in the revised manuscript (Line248-250). The data frame dimension is 14. We really appreciate your comments, and we have provided a flow chart of data preprocessing in the supplementary figure as your suggestion. It is shown as follows:

We mixed the sample points of the model group (isoflurane anesthesia and surgery) and the Dex-treated group (Dexmedetomidine and isoflurane anesthesia and surgery) for classification, trying to find the similar trend between them. In the classification

process, through the minimum classification error and confusion matrix, we show that pod and non-POD are well separated, and in the process of low-dimensional space visualization, POD and non-POD are well separated, which reflects the interpretability of this model to a certain extent.

Since we apply the same mapping rule to the Dex-treated group and the model group, which means that our criteria are consistent, it is reasonable to assume that the sample point of the Dex-treated group with a close distance to pod is Dex-treated-POD, and that of the Dex-treated group with a close distance to non-POD is Dex-treated-non-POD.

Q5: In my opinion, a table defining the n for each experiment will help the reader to understand the methodology better.

A5: Thank you for your suggestion. We provide a table with sample sizes of each experiment in the supplementary figures section.

Q6: In figures 2c, 3b, 3s b&c, and 5s a&b, the figure legends do not provide enough information. There is some much of an overlap that, in my opinion, different means of showing these data are needed.

A6: Thank you for your suggestion. We have removed the duplicated parts according to your suggestion. We have made changes to the figures and legends based on your comments.

Q7: In figure 5s, the confusion matrix data suggest perfect model performance. At the same time, in the same figure panel d some mice converted from sick to healthy and vice versa at different points after surgery, suggesting mistakes in model definitions.

A7: We are sorry for the inappropriate expression of this part that make you confused to our result. According to your comments, we revised the Supplement figure5 in the revised manuscript. As follow:

In addition, we explain your confusion as follow:

First, the training set of confusion matrix was the 14 significant different movements from POD group and non-POD group. And the test set is these 14 movements of the 8 Dex-treated mice in four days. The detection was repeated four times in four days for each mouse, in other words, the sample size of test set is 32. We performed four days because the onset time and recovery time are not uniform, which is an important feature of POD.

In supplement figure5, panel c is showing the state of test set in four days (POD or non-POD), there are some mice from sick to healthy and vice versa at different points after surgery. The confusion matrix is derived from the cross-validation results. There is no conflict with the classification error of the training set shown by the confusion matrix. To avoid confusing the reader, we have modified the figure in the revised manuscript.

Q8: In the discussion section, the authors state: "... limitations of this study include the failure of the authors to cite enough literature on some behaviors...". If the authors think that more literature needs to be cited, what stopped them from doing so?

A8: We are sorry for the wrong expression of this sentence. We have rewritten the sentence "the limitations of this study include there are not enough literatures about the meaning of spontaneous behaviour, which can be attributed to the limited animal behavioural repositories. This can be addressed through the continuous carrying out of animal behaviour studies" (Line 579-581). In addition, we explain your confusion as follow:

In our manuscript we find a lot of behaviors in POD mice that are statistically different from non-POD mice. Based on the observations to video clips of movements, we name them, e.g., rear with exploration. However, in the pursuit of the biological significance of these movements, limited by the relatively few reports on the behaviour of animals, we can only interpret the behavioral significance of these different movements through these existing reports.

Reviewer #2 (Remarks to the Author):

In the manuscript the authors based on recent advanced animal behavior recording and analyzing techniques to present a multi-scaled behavioral phenotyping framework for disease animal model assessment. The authors used several aspects of behavioral feature to demonstrate the benefits of comprehensive behavioral profiling method for the identification of animal diseases, classification of disease subtypes, and evaluation of drug efficacy etc. Most significantly, the quantifiable behavioral data presented in this study are consistent with symptoms of the disease in other species, especially humans.

This experiments of this work was well designed, the data analysis and visualization were novel, and the manuscript was well organized. I believe this work will inspire more researchers in this fields to characterize the behaviors of animal disease model across-species. Before this manuscript could be accepted for publication in the Communications Biology, there are minor flaws that need to be improved. Here below a few comments that I hope will help to improve this work.

1) The novelty of this study is to emphasis the multi-scale, hierarchical and dynamic behavioral properties, which can better represent the disease symptoms of animals compared with the traditional static behavioral parameters. Therefore, in the second paragraph, the authors introduced that traditional behavioral assessment methods mainly use the behavior tests such as EPM, OFT, and NOR, by measuring static parameters, such as the time an animal spends performing a specific task in a confined chamber. Then In the third paragraph, the authors cited many literatures, explaining that animal behavior should be hierarchical, dynamic, time-varying, which has the human language-like characteristics. And explains why animal behavior has this characteristic from the neural-behavioral correlation relationship aspect.

My suggestion for the third paragraph is that there needs a clearer explanation of why multi-scale, hierarchical, dynamic behavioral features are better for evaluating POD. It

needs to be elaborated from the relationship between neural activity, symptoms, and behavior. I recommend first explaining the relationship between disease symptoms and abnormal neural activity, then demonstrating that symptoms are reflected in multi-aspects behavioral characteristics, and finally emphasizing that animal behavior is shown to be such a multi-level structure, thus spontaneous behavior, task-driven behavior is necessary for POD assessment.

A: Thank Reviewer 2 for the positive comments and constructive suggestions to improve our paper. According to your suggestion, we have changed the expression of this part (Line65-82).

2) Line 286: “we labeled the 3D skeleton information of the mice using DeepLabCut (DLC) which included 16 body parts of the mice with 3D coordinates”.

A: Thank you for your suggestion. According to your suggestion, we have added this part in the revised manuscript (Line 157-163).

I think here should be label 2D pose for four single views, not 3D skeleton. The 3D coordinates are reconstructed from four 2D coordinates. Therefore, the process of this part should be rewrite and add these details. Accordingly, in Fig. 1c, the other 3 views of the DLC pose estimation images need to be added.

A: Thank you for your suggestion. According to your suggestion, we have changed the flow chart.

3) Line 431: 11 postural features do not seem to correspond to Fig. S4a-m, 13 postural features? Besides, “eigenvalue” should be replaced with “features”. Because eigenvalue is a term for linear algebra, not for certain behavioral parameters of an animal.

Thank you for your suggestion. We apologize for the wrong writing and have corrected it in the revised manuscript. Here we used “eigenvalue” to describe the 7 values extract from the PMF. To avoid the misunderstanding, we revised this part in the revised manuscript (Line185-188).

4)Both spellings of “behaviour” and “behavior” exist in manuscripts, the spelling needs to be improved.

Q: We are sorry for our carelessness and we will unify the spelling of the words in the revised manuscript.

Although the authors present a lot of data and I think their motivation for characterizing behaviors in rodents post-anesthesia might be good, there are too many things wrong with this manuscript to describe in detail. This review hits the highlights, however even if these issues were remediated the other glaring errors might become even more apparent.

A: Thank you for your approval of the experimental content and methods. We will try

our best to answer your suggestions.

Q1: First, the logical flow of the paper is in question. As best I can tell the authors intend to use AI and video data from multiple cameras to demonstrate efficacy of dexmedetomidine in prevention or treating delirium in their rodent model. This is “the cart before the horse”.

A1: Thank you for your advice. Firstly, dexmedetomidine is one of the most promising drugs for POD prevention. The dexmedetomidine dose we selected was based on a survey of previous reports. They were proved that dexmedetomidine at 25ug/kg had a good neuroprotective effect. Therefore, in this study, we made a detailed comparison of the effect of dexmedetomidine on postoperative behaviors, including postoperative spontaneous behaviors and cognitive-related task-driven behaviors. The results showed that dexmedetomidine did have a significant effect on some movements. This may be one of the clues in the future to explain the mechanism of dexmedetomidine in preventing delirium. Secondly, the evaluation of the effects of dexmedetomidine was an exploration of the effects of Dex in our study, not an attempt to validate the findings that POD mice produce abnormal behaviors in spontaneous and task-driven behaviors.

Q2: First, there is no standard rodent model for post-operative delirium. I think the authors are attempting to propose that their methodology can detect a rodent version of post-operative delirium, but it is not explicitly stated like that. The text of the Results section cannot be followed by the reader.

A2: Thank you for your suggestion and affirmation. As you mentioned, there is no gold standard for the diagnosis of POD in rodents at present. It is undeniable that POD has been discussed endlessly, and the research on POD mechanism is also an important issue in the field of anesthesia. We found that previous studies used the center time of open-field tests to evaluate POD. Center time uses the habits of rodents as a classic behavioral indicator of anxiety. Whether it could exhibit the complex manifestations of POD attracted our attention.

Since there is no gold standard for animal behavior in POD, we used unsupervised clustering to distinguish between POD and non-POD groups by similarity to mice with normal behavior (control group). And we found significant differences in spontaneous and task-driven behaviour between POD and non-POD groups. In addition, we showed that there were indeed cognitive changes in the POD group with the cognitive assessment method reported in *Cell*[2].

We believe that our detailed evaluation of spontaneous behaviors and task-driven behaviors in animals after surgery is of great significance to the basic research of POD and will be used by other researchers in the future.

Sorry for giving you a bad reading experience, we have rewritten the results section to make it easier to understand in the revised manuscript.

Q3: The authors state that “delirium-like behaviors” (not described at all) are able to be detected with their methodology and seven separate behaviors obtained from their methods (these seven not well described) demonstrate rodent delirium. In order to legitimately make the claim that these behavioral tests that “diagnose” delirium in mice the authors would have to show more commonality with human delirium. For example are these abnormal recoveries more likely in aged animals or long surgeries?

A3: Sorry for the unclear description, we have made a more detail description in the revised manuscript (Line517-529). In our study, to increase the credibility of the experimental results, we also evaluated the traditional behavior of mice, which was consistent with the previously reported behavioral results of mice. And for clinical application, we also found that in our model, the POD mice met the characteristics of CAM scale: early onset, fluctuating state, cognitive decline, psychomotor changes, attention decline, etc. Therefore, we believe that in our study, partial clinical features of POD were reproduced in mice using spontaneous and fine behaviors.

Q4: The figure doesn't shed light on the situation, why does left turning predict delirium and not right-turning? What does specifically looking to the right have to do with the mechanism of delirium?

A4: In our research, only the left turning was statistically different between two groups, not the right turning. Moreover, statistical significance sometimes does not imply biological significance, and it may be that the biological significance of the action needs further discovery and interpretation, or it may not be, as described in limitation (Line579-581). Our future studies will therefore focus on these movements that are both differential and interpretable.

Q5: The videos reference are so short (some less than 1 second)– there is no context for any of these behaviors.

A5: We recorded 10 min video of mice during OFT and NOR with 30fps. The supplementary video showed 40 video clips of decomposition behavior. Due to the file size limitation of the submission system, we could not upload the original video, so we compressed the clarity (150MB to 5MB) to ensure a smooth upload (Supplement video21-24). They are videos from four perspectives of the same experiment, and we labeled the movement number to each frame (Supplement Table 2).

Q6: I have read the results section several times but I cannot understand how the authors progress from these associations to a “model” group and a red/green group which is

some sort of group intended to represent a cohort with an expected delirium rate? Maybe. The description is hard to follow.

A6: Thank you for your efforts in our research. According to your comments, we have changed the figures, legends and descriptions of results in this part. The model group was mice that underwent anesthesia and surgery, and the control group was mice that did not undergo anesthesia and surgery. Red group or green group was the subcluster of the cluster analysis. They were not equal to the model and control groups. In the modified figure, we changed the presentation of this section. As follow:

Q7: Additionally, the fact that dexmedetomidine decreases these “abnormal” behaviors does not “prove” these behaviors are delirium. Although delirium rates might be lowered with dexmedetomidine it is possible that in this model, dexmedetomidine is causing an overall increase in quiescence. Overall activity level is not really controlled for in the behaviours indicative of rodent delirium.

A7: According to your suggestion, we evaluated the effect of dexmedetomidine on postoperative velocity, and the results showed that there was no significant difference between the two groups.

Q8: Second, in multiple places throughout the paper the language is very confusing and non-standard for rodent researchers in the area. The concepts of “model group” and “control group” and red/green designations are not clear at all.

A8: Thank you for your comments. We have made modifications to the naming problem in the revised manuscript, and described the experimental grouping in detail in the methods section (Line110-114). In addition, in order to better display the results, we modified Figure2 in the revised manuscript. We have changed the representation of clustering results in the hope that this can better display our results.

Q9: Additionally, I am not sure what the authors mean when they write “static parameters” for behavior?

A9: Sorry for the unclear description to “static parameters”. Here we try to use static parameters to refer to the traditional behavior of center time, open arm time, total distance and so on. Their common characteristic is that they ignore the inherent dynamics of the behavior itself. In other words, previous studies only looked at how long the mice were in the center, not what they were doing. In the abstract, we do not explain the static parameters in detail due to the word limit. But we talked about it in detail in the introduction (Line 56-64). If this still confuses you, we can change the wording, but in the revised manuscript we still use 'static parameters' to describe the general characteristics of traditional behaviour.

Q10: While some of this may be due to translation services being used (were translation services used?). It sounds a little like the kind of word salad that come from AI technology. Most of the key points were wordy and awkwardly described.

Reviewing the text of this manuscript was arduous. Obvious examples of this are in the abstract where the authors describe delirium as “pernicious”. While, delirium is often underdiagnosed and can lead to an escalation of care, it is (itself) not progressive per se but rather associated with the development of cognitive decline in the long-term, such as an exacerbation Alzheimer’s Disease and related dementias.

A10: We are sorry for bringing you a bad experience during reviewing our manuscript. As you said, this manuscript has been professionally polished. According to your suggestions, we have made modifications in the revised manuscript.

The next sentence of the abstract is arguably even harder to understand. “Traditional behavior analysis mostly focuses on static parameters, which lost the structure of animal behavior, or is based on limited observations of the spatial positions of the mice and their derivatives.” It has awkward syntax, a verb conjugation error, and confusing. I think the authors are trying to say that: Most rodent behavioral analysis programs focus on identifying particular states of behavior that occur intermittently (i.e., grooming or freezing) and does not incorporate temporal dynamics or multiple camera angles.

A: Sorry for making you confused by this sentence. In view of the word limitation in the abstract section, we used 'static parameters' to summarize the previous behavioral

detection methods of POD. We rewrote the sentence in the abstract of revised manuscript (“Traditional behaviour analysis mostly focused on static parameters. They are based on the limited observations of mice’s positions particular states, which lose the structure of behaviours”) (Line24-25). In the introduction, we expand on the characteristics of traditional behaviour (Line 56-64). Static parameters correspond to our elaboration in the next paragraph that behaviour itself is structured and is fluid (Line 68-75).

Q11: Honestly, I am not really sure what they are attempting to convey, but I provide my interpretation as to their meaning based on how I would motivate this project if it were my data.

A11: We sincerely thank you for your comments and suggests to our research. We have tried our best to answer your questions and made modifications according to your suggestions.

1. Nagai, J., et al., *Hyperactivity with Disrupted Attention by Activation of an Astrocyte Synaptogenic Cue*. Cell, 2019. **177**(5).
2. Shan, W., et al., *Critical role of UQCRC1 in embryo survival, brain ischemic tolerance and normal cognition in mice*. Cellular and Molecular Life Sciences : CMLS, 2019. **76**(7): p. 1381-1396.